# Host-pathogen interactions in the *Plasmodium*-infected mouse liver at spatial and single-cell resolution

Franziska Hildebrandt [1] ✉, Miren Urrutia Iturritza [1], Christian Zwicker[2,3], Bavo Vanneste[2,3,4], Noémi Van Hul [5], Elisa Semle[1], Jaclyn Quin[1], Tales Pascini[6], Sami Saarenpää [7], Mengxiao He [7], Emma R. Andersson [5], Charlotte L. Scott [2,3], Joel Vega-Rodriguez [6], Joakim Lundeberg [7] & Johan Ankarklev [1] ✉

Upon infecting its vertebrate host, the malaria parasite initially invades the liver where it undergoes massive replication, whilst remaining clinically silent. The coordination of host responses across the complex liver tissue during malaria infection remains unexplored. Here, we perform spatial transcriptomics in combination with single-nuclei RNA sequencing over multiple time points to delineate host-pathogen interactions across *Plasmodium berghei*-infected liver tissues. Our data reveals significant changes in spatial gene expression in the malaria-infected tissues. These include changes related to lipid metabolism in the proximity to sites of *Plasmodium* infection, distinct inflammation programs between lobular zones, and regions with enrichment of different inflammatory cells, which we term 'inflammatory hotspots'. We also observe significant upregulation of genes involved in inflammation in the control liver tissues of mice injected with mosquito salivary gland components. However, this response is considerably delayed compared to that observed in *P. berghei*-infected mice. Our study establishes a benchmark for investigating transcriptome changes during host-parasite interactions in tissues, it provides informative insights regarding in vivo study design linked to infection and offers a useful tool for the discovery and validation of de novo intervention strategies aimed at malaria liver stage infection.

Infectious *Plasmodium spp.* sporozoites, transmitted by female *Anopheles* mosquitoes, are deposited in the dermis after a mosquito bite, disseminate through the circulation, and can eventually infect a liver hepatocyte[1]. Inside the hepatocyte, the parasite resides within a parasitophorous vacuole (PV), surrounded by a parasitophorous vacuole membrane (PVM), which is formed by invagination of the host cell membrane and assists it in evading detection by the immune system. Moreover, *Plasmodium* can exploit this interaction surface to

[1]Molecular Biosciences, the Wenner Gren Institute, Stockholm University, Svante Arrhenius Väg 20C, SE-106 91, Stockholm, Sweden. [2]Department of Biomedical Molecular Biology, Faculty of Sciences, Ghent University, Ghent, Belgium. [3]Laboratory of Myeloid Cell Biology in Tissue Damage and Inflammation, VIB-UGent Center for Inflammation Research, Technologiepark-Zwijnaarde 71, Ghent 9052, Belgium. [4]Laboratory of Myeloid Cell Biology in Tissue Homeostasis and Regeneration, VIB-UGent Center for Inflammation Research, Technologiepark-Zwijnaarde 71, Ghent 9052, Belgium. [5]Department of Cell and Molecular Biology, Karolinska Institutet Stockholm, SE-171 77, Solna, Sweden. [6]Laboratory of Malaria and Vector Research, National Institute of Allergy and Infectious Diseases, National Institutes of Health, 12735 Twinbrook Parkway, Rm 2E20A, Rockville, MD 20852, USA. [7]SciLifeLab, Department of Gene Technology, KTH Royal Institute of Technology, Tomtebodavägen 23a, SE-171 65, Solna, Sweden. ✉e-mail: franziska.hildebrandt@su.se; johan.ankarklev@su.se

alter the hepatocyte and access nutrients or other essentials required for its growth and development[2,3]. The parasite then transitions into the symptomatic blood-stage by releasing thousands of merozoites[4] from an infected hepatocyte, at around 48 h post infection (hpi) for the rodent-specific *Plasmodium berghei* parasite[5]. Notably, the liver represents a major bottleneck during the malaria life cycle and is the stage targeted by the only two WHO-recommended malaria vaccines to date. Despite the limited efficacy (36% in children 5–17 months of age[6,7]) of both the RTS,S and R21/Matrix-M vaccines, the pre-erythrocytic stages of malaria infection show substantial promise for further vaccine development.

The liver serves as a critical immune organ, detecting and eliminating pathogens and toxins while simultaneously regulating energy, lipid, and protein synthesis[8,9]. Its structural organization consists of lobules. Each lobule describes a hexagonal unit with portal veins at the corners and a central vein at the center. To ensure optimal metabolic activity and to prevent futile cycles, liver cells express differential proteins along the axis from each portal node to the central vein, constituting spatially defined metabolic zones, commonly referred to as zonation[10,11]. Labor is further divided amongst the highly diverse cell types of the liver, including parenchymal cells, such as hepatocytes and cholangiocytes, which account for 70 – 80% of the total liver area, as well as non-parenchymal cells (NPCs). NPCs include liver sinusoidal endothelial cells (LSECs), which line the vasculature of the liver, as well as Kupffer cells and other immune cells, including neutrophils, T and B lymphocytes, natural killer (NK) cells, and NKT cells, which are found scattered across hepatic lobules[12,13]. The portal vein is considered the main entry point of gut-derived pathogens making this position in the liver more susceptible but uniquely equipped to respond to circulating pathogens[8,13]. Maintaining immune balance is crucial for liver function, as disturbed homeostasis or prolonged inflammation can lead to severe diseases like cirrhosis, non-alcoholic steatohepatitis, hepatocellular carcinoma, and liver failure[14]. However, pathogens like *Plasmodium* may exploit the liver's immune tolerance[15].

During infection, *P. berghei* elicits a sequential transcriptional response in the liver of the host, including interferon-mediated immune genes expressed at later parasite developmental stages[16–18]. Parasite development in the liver is heterogeneous and suggested to be affected by zonation, where abortive infections in periportal zones have been described[19]. These findings have advanced our understanding of *Plasmodium* infection and hepatocyte zonation, as well as tissue-wide immune responses. However, a comprehensive spatial map of host–parasite interactions, including gene expression profiles in their true tissue context, beyond hepatocyte zonation, and including the involvement of liver resident immune cells, has been missing.

In our previous work, we established the first spatial transcriptomics map of murine liver tissue, including expression-by-distance measurements of target structures[20]. Here, we perform spatial gene expression analysis of *P. berghei*-infected mouse livers over multiple time points during infection (12, 24, and 38 h post infection (hpi)) to map out genes and genetic pathways involved in host–parasite interactions across liver tissues. In this study, we use a combination of the original Spatial Transcriptomics 2K arrays[20,21] (henceforth referred to as ST) and Visium arrays (10X Genomics Inc.)[22]. Spatial data resulting from ST enabled us to investigate a large sample size (*n* = 38 tissue sections), whereas the Visium arrays (*n* = 8 tissue sections) allowed for increased resolution of expression analysis due to the decreased spot size (55 vs. 100 μm) and shorter distances between spot-centers[21]. Additionally, we performed single-nuclei RNA sequencing (snRNA-seq) on the same tissue samples to identify and deconvolve cell types. This integrated approach allows for a comprehensive transcriptomics analysis of *P. berghei*-infected liver sections, including complete cell type information.

Combining spatial transcriptomic and snRNA-seq data reveals both global and local effects of *P. berghei* infection compared to controls in liver tissue. Notably, we identify differential expression of genes involved in lipid homeostasis at infection sites, potentially indicating a parasite immune evasion strategy. We also uncover unique tissue structures termed inflammatory hotspots (IHSs) that are both morphologically and transcriptionally distinct from surrounding tissue, and resemble foci of immune cell infiltrates, as observed in liver pathologies of various diseases[23–25]. We validated our transcriptional analysis by establishing that these IHSs are enriched in immune cell infiltrates, leading us to propose that they are a distinct spatial feature of the immune response to *P. berghei* infection. In total, this study provides a highly informative resource on spatiotemporal host tissue responses during malaria infection and development in the liver.

## Results

### Spatial transcriptomics captures liver tissue responses induced by malaria parasite infection

We used Spatial Transcriptomics (ST)[20] to analyze 38 liver sections infected with either *P. berghei* parasites or injected with *A. stephensi* salivary gland lysate (SGC) at different time points (12, 24, and 38 h post injection). We added Visium Spatial Gene Expression analysis of eight additional liver sections for higher spatial resolution (see Methods for details), resulting in a total of 46 spatially analyzed liver sections collected from 18 adult female mice. The comparison with SGC sections enabled us to control for mosquito-related responses. Each tissue region that forms a data point in the ST analysis consists of a small mixture of cells[20]. Therefore, we performed additional single-nuclei RNA sequencing (snRNA-seq) to deconvolve spatial data and artificially increase the resolution in our analyses (Fig. 1a).

We first identified spatial expression patterns related to infection by performing unsupervised clustering analysis, which groups data points based on their similarity in gene expression profiles (see Methods for details). We identified 12 clusters for the ST data (ST1-ST12) (Fig. 1b and Supplementary Figs. 1–3) and 10 clusters for the 10X Visium data (V1–V10) (Supplementary Fig. 6). Many of these clusters (ST1, ST4-ST5, and ST7-ST8) represent the spatially different patterns of gene expression in healthy liver tissue, and have previously been described[20]. Four of these ST clusters—namely ST3, ST10, ST11, and ST12—exhibited a unique pattern of gene expression influenced by the condition, i.e., *P. berghei* infection or SGC challenge, and the collection time point (12, 24, or 38 h) (Fig. 1b and Supplementary Fig. 4). At 12 hpi with *P. berghei*, a large proportion of spots displayed an expression profile associated with cluster ST3, while SGC-challenged mice did not show an increased proportion of spots displaying an expression profile associated with cluster ST3 until later time points. We made a similar observation for cluster ST10, but with fewer associated spots. Spots belonging to cluster ST11 showed enrichment in sections infected with *P. berghei* parasites, while spots of cluster ST12 gene expression were missing entirely from the SGC sections (Fig. 1b and Supplementary Fig. 4).

Differential gene expression analysis (DGEA) revealed that cluster ST12 is defined by upregulation of *P. berghei*-specific transcripts (*HSP70-pb*, *HSP90-pb*, and *LISP2-pb*), suggesting they represent parasite-infected tissue sites (Supplementary Fig. 5 and Supplementary Data 1). Spots associated with clusters ST10 and ST11 exhibit an anti-correlated presence along the infection timeline. Further, DGEA and gene ontology (GO) enrichment suggest that cluster ST10, predominantly observed during early infection, is associated with pro-inflammatory signaling (e.g., IL-17 and TNF pathways), including phagocytosis, and KEGG-terms including leishmaniasis and tuberculosis. In contrast, cluster ST11 shows gene set enrichment of pathways related to intracellular pathogen signaling (NOD-like and RIG-I-like receptor pathways), complement and coagulation cascades, and KEGG-terms associated with viral infections such as COVID-19 and hepatitis C. Moreover, most upregulated genes in cluster ST11 are

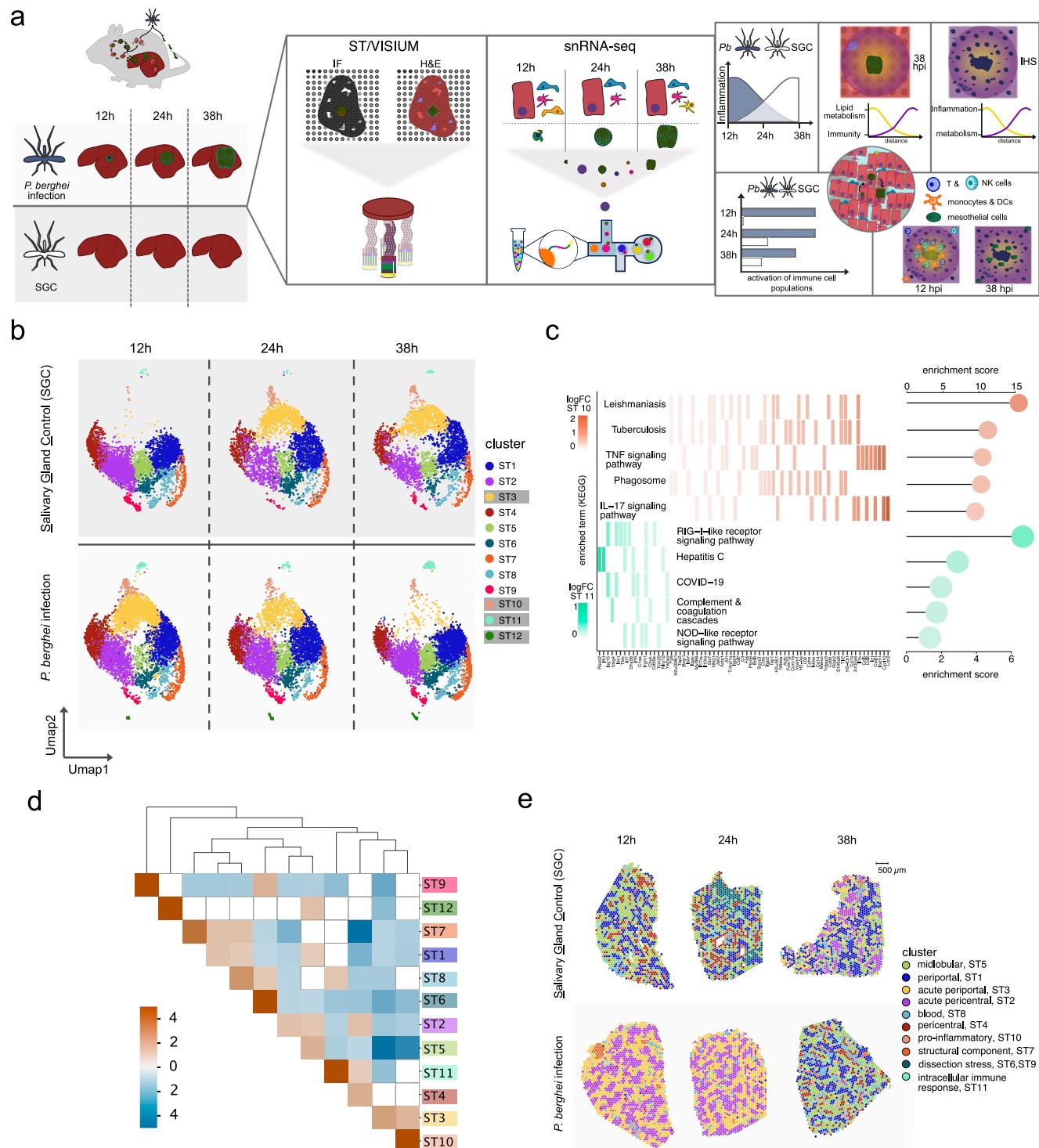

interferon-stimulated genes (ISGs), including *Ifit1*, *Ifih1*, *Irf7*, and *Irf9* (Fig. 1c, Supplementary Fig. 5, and Supplementary Data 1).

Cluster ST3 exhibits upregulation of genes linked to acute phase response and inflammation, including the *Saa*[26,27] and *Orm* families[28] (Supplementary Fig. 5 and Supplementary Data 1). The higher prevalence of cluster ST3 associated spots at 12 hpi suggests an initial inflammatory stress response in the *P. berghei*- infected liver, which is delayed in the SGC sections.

In addition, we identified three clusters (ST2, ST6, and ST9) with previously undescribed expression profiles. These clusters do not show clear links to *P. berghei* infection or SGC challenge (Fig. 1b and Supplementary Data 1). Cluster ST2 exhibits expression of a number of

genes which are associated with pericentral localization, such as *Cyp2e1*[11,20] (Supplementary Fig. 5), suggesting it may represent an intermediate zone between central and portal areas, closer to the central region. We confirmed this by analyzing cluster interactions, showing that spots of cluster ST2 are enriched adjacent to spots of cluster ST4 (Fig. 1e), supporting its pericentral proximity.

Comparing ST (ST1–ST12) and Visium (V1–V10) data reveals high overlap between differentially expressed genes (DEGs) across identified clusters (Supplementary Fig. 6 and Supplementary Data 1). Notably, spots associated with *P. berghei* infection (cluster ST12) are not present in every analyzed infected tissue section. This is especially the case at the early infection time points where the number of detected *P.*

**Fig. 1 | Spatial organization of livers infected with *P. berghei* parasites or salivary gland lysate (SGC). a** Schematic representation of the experimental design of this study. Livers were collected at 12, 24, or 38 h post infection (hpi) with *P. berghei* parasites or salivary gland lysate of uninfected mosquitoes (SGC) (left). Immunofluorescence (IF) staining of the parasite, spatial transcriptomics (ST) or 10X visium spatial technology protocols, and droplet-based single-nuclei RNA sequencing (snRNA-seq) were performed (center). Both data were further analyzed computationally (right). IHS stands for immune hotspot. **b** After dimensionality reduction, the normalized and batch-corrected data were embedded in UMAP space and split by the original condition for visualization. Data from SGC sections are shown on the top from 12 to 38 hpi (left to right) and data from *P. berghei*- infected sections are shown on the bottom from 12 to 38 hpi (left to right). Clusters with an obvious association to infection conditions are highlighted with gray boxes. **c** For identified clusters ST10 and ST11, differential gene expression analysis (DGEA) was performed, followed by functional enrichment analysis for each cluster (see Methods for details). Overrepresented pathways of the KEGG database for ST10 are shown in

rose and for ST11 in aquamarine. Scales for expression values of overrepresented genes belonging to the individual KEGG pathways are shown for ST11 (left) or ST10 (right), from high expression (dark) to lower expression (light). Selected gene names are shown at the bottom. Enrichment scores for the pathways are shown on the right. **d** Interaction analysis of clusters was performed to evaluate spatial enrichment expression programs as suggested by clustering analysis in space. Positive enrichment values (orange) indicate spots belonging to these clusters are more likely to be neighboring, while negative enrichment values (blue) indicate spots associated with these expression programs are less likely to be neighboring. Clusters without significant enrichment in each other's neighborhoods are shown in white. **e** Visium clusters were imposed on spatial positions and annotated according to spatial expression features. Sections of the investigated conditions are divided for ease of inspection as in (**b**), with the top panel comprising SGC sections across 12–38 hpi and the bottom panel comprising *P. berghei* infected sections across 12–38 hpi.

*berghei* transcripts is lower, emphasizing the value of larger sample sizes for ST experiments. Interestingly, at the higher resolution of Visium, we see a distinction of spatial gene expression patterns in clusters ST2 (acute pericentral) and ST3 (acute periportal) (Fig. 1c), which suggests zonation of the acute phase and inflammatory response (ST3) to periportal regions during infection.

## *P. berghei* infection impacts both proximal and peripheral gene expression in liver tissue

We found the majority of uniquely DEGs between *P. berghei* infected and SGC sections at 12 and at 38 hpi (Supplementary Fig. 7 and Supplementary Data 2). Upregulated genes at 12 hpi in *P. berghei*-infected tissues are linked to cellular stress responses, including transcription of *Saa1*, *Saa2*, *Saa3*, and *Lcn2*[27,29]. Meanwhile, most upregulated genes at 38 hpi are ISGs, including *Ifit1*, *Ifit3*, *Irf7*, and *Usp18*, which have been previously implicated with an interferon response towards *Plasmodium* liver infection[16,19,30] (Fig. 2a and Supplementary Data 2).

Modules of stress response genes at 12 hpi and ISGs at 38 hpi exhibited higher expression in infected sections, but this expression was not confined to the infection sites, suggesting a widespread inflammatory response across the tissue (Fig. 2b). Cluster ST11 displayed the highest expression of ISGs, indicating that the locations of cluster ST11 represent foci of a type I IFN response (Fig. 2c).

Unsupervised clustering results indicate parasite localization across the infected tissues. However, determining parasite positions solely at the RNA level proves challenging due to limited spatial resolution and low parasite transcript abundance. Despite these challenges, we can detect an increased number of parasite transcripts in the infected conditions over time (Supplementary Fig. 8). In addition, robust validation of parasite positions and development is achieved through immunofluorescence (IF) staining using the parasitophorous vacuole membrane (PVM) marker UIS4 (Fig. 2d).

We performed a correlation analysis between the distance to the neighborhood of the parasite and gene expression (see Methods for details). To facilitate the interpretation of expression changes (Δ) across conditions, we centered expression at 0 μm from the parasite neighborhood. A negative correlation signifies reduced expression with increased distance to the parasite, while positive values indicate increased expression with increased proximity to the parasite. We observed significantly reduced parasite gene expression with an increased distance to parasites (within 400 μm of parasite neighborhoods). The strongest changes in expression as a function of the distance to parasite neighborhoods were observed at 38 hpi (Fig. 2e).

Despite the significantly lower abundance of parasite transcripts compared to the host, we performed DGEA in parasite neighborhoods, aligning it with Afriat et al.'s pseudotime analysis[19]. This revealed that a high proportion of genes from our data are linked to early latent time determined by Afriat et al., which can possibly be explained by the

sparse presence of *P. berghei* transcripts in our data (Supplementary Fig. 9).

Next, we determined host gene expression with positive and negative correlation to *P. berghei* infection sites across all time points and performed a GO-term enrichment analysis (see Methods for details). The GO-term enrichment indicates higher expression of genes involved in the chemotaxis of leukocytes, including expression of *Xcl1*, *Fcer1g*, and *Csf1r* near the parasite at 12 and 24 hpi. However, the pattern is reversed at 38 hpi, with decreased expression of the genes described above, along with other genes, including, *Msr1*, *Cd74*, *Csf3r*, and *Camk1d*, in proximity to the parasite (Fig. 2f, g, Supplementary Figs. 10–13, and Supplementary Data 3).

Leukocyte chemotaxis is crucial for inflammation and immune responses and includes the recruitment of macrophages and neutrophils to ward off invading pathogens[31,32]. Our data suggest that the parasite triggers a pro-inflammatory response near the infection site but evades phagocytosis during the late infection time point, just prior to egress from the liver. Notably, at 38 hpi, gene expression values such as *Msr1* and *Cd74*, which are associated with inflammation[33,34], showed a positive correlation with increasing distance from the parasite (Fig. 2g, Supplementary Fig. 12, and Supplementary Data 3). Additionally, we found *Insig1*, which is linked to lipid homeostasis and the prevention of lipid toxicity[35], to positively correlate with increasing distance to parasite neighborhoods (Fig. 2g, Supplementary Fig. 12, and Supplementary Data 3).

In the proximity of parasite locations, we observed higher expression of *Fabp5*, involved in the regulation of lipid metabolism, peroxisome proliferator-activated receptors (PPARs) and cell growth[36,37]. We also identified a higher expression of *Mospd2*, implicated in host-pathogen interactions with *T. gondii*[38], and a higher expression of *Rheb*, which activates mTORC1, promoting proliferation and survival. Moreover, Rheb is shown to be involved in the limitation of autophagy[39–41], an increasingly recognized pathway in *Plasmodium* liver infection[42,43] (Fig. 2h, Supplementary Fig. 13, and Supplementary Data 3).

## Inflammation exhibits spatial patterns in response to *P. berghei* and SGC challenge

Several studies report that parasite localization in the different metabolic zones of the liver influences the developmental progress of *Plasmodium* in hepatocytes, and suggest higher developmental success in pericentral areas[19,44,45]. While our data do not show a linear correlation between hepatic zonation and *P. berghei* localization in liver tissue, we observe similar trends, where parasite gene expression is higher in areas within 400 μm of computationally annotated pericentral veins (see Methods for details). In addition, our data suggest that a large proportion of parasites are present and transcriptionally active in areas that we defined as intermediate, situated beyond

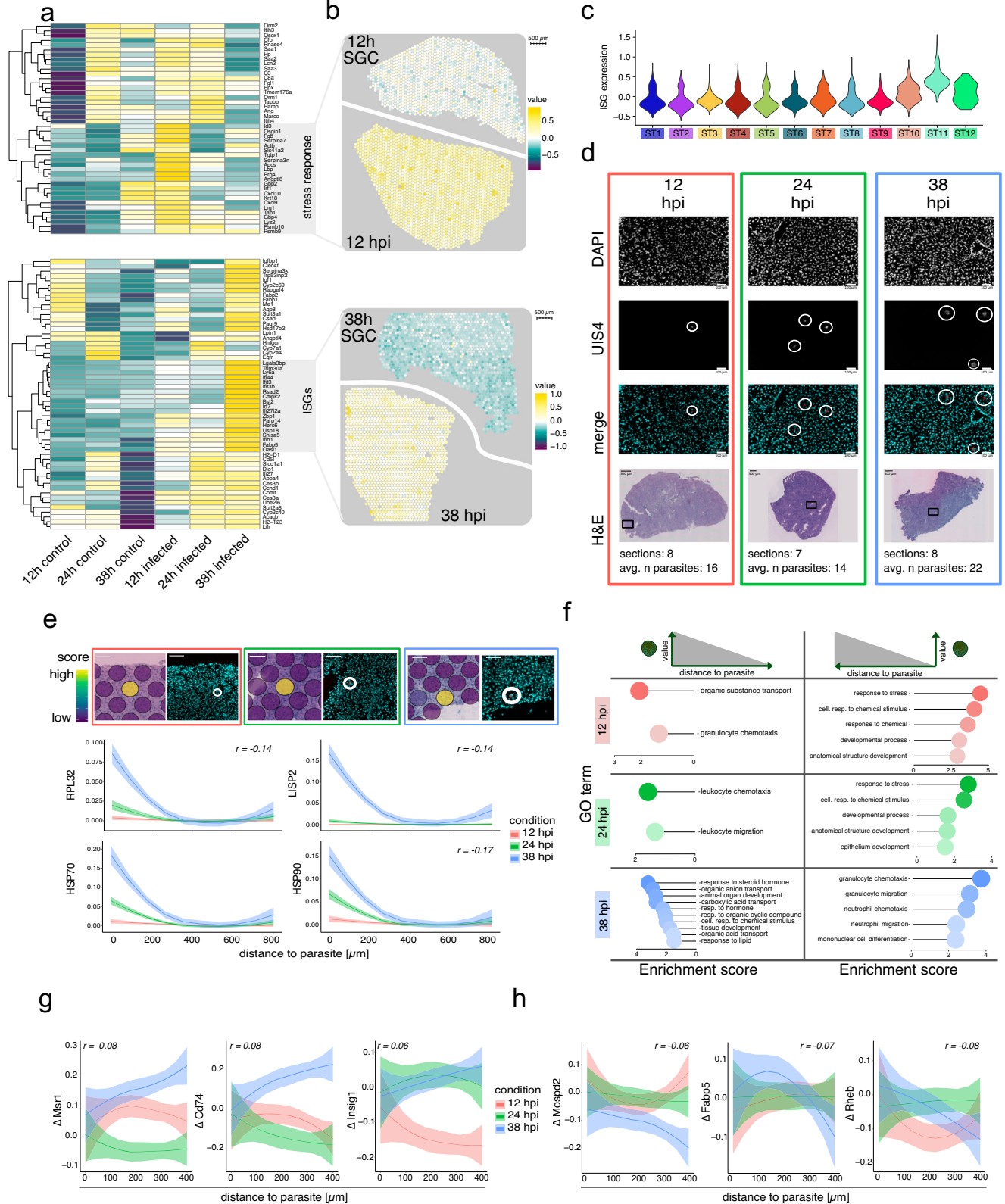

400 µm from both pericentral and periportal neighborhoods (Supplementary Fig. 14).

Our data further suggest hepatic zonation of inflammatory responses at 12 and 24 hpi (Fig. 1e). To further validate this observation, we investigated correlations between periportal marker genes (*Cyp2f2, Sds*), pericentral marker genes (*Glul, Slc1a2*) and differentially expressed genes in the acute periportal cluster (ST3) or the acute

pericentral cluster (ST2). Marker genes of ST2 (*Car3, Ces3a, Ces1d, Cyp3a11*, and *Nr1i3*) correlate with gene expression of pericentral marker genes while marker genes of ST3 (*Itih3, Itih4, C3, Ambp, Fgg, Qsox1*, and *Hpx*) correlated with periportal marker genes (Fig. 3a), supporting the notion of hepatic zonation of these clusters. Expression-by-distance analysis further validated zonated expression profiles of ST3 acute periportal and ST2 acute pericentral genes

**Fig. 2 | Global and spatially distinct effects of *P. berghei* on tissue gene expression. a** Differentially expressed genes between *P. berghei* infected and SGC sections at 12 hpi and 38 hpi. Averaged expression is depicted from low (dark purple) to high (yellow) expression. **b** Expression of modules showing highest expression in 12 hpi and 38 hpi across tissue spots for infected and SGC sections. The scale bar denotes 500 μm and module values range from low (dark purple) to high values (yellow). **c** Expression of ISG module expression across spatial clusters. **d** Immunofluorescence (IF) and Hematoxylin and Eosin-stained (H&E) images of *P. berghei* infected tissue sections at 12, 24, and 38 h (*n* = 23). Colored boxes indicate time points (12 hpi = red, 24 hpi = green, 38 hpi = blue). Parasites (UIS4) are highlighted by white circles, scale bars indicate 100 μm and DAPI denotes DNA staining. The position of each parasite is indicated in black on H&E images. The number of sections per time point and average number of parasites per section is shown below. **e** Loess-smoothed module scores of *P. berghei* genes with negative correlation (*r*) to parasite distance. Ribbons around the curve indicate the standard error of the mean (SEM). Colors indicate conditions (12, 24, and 38 hpi) and white circles highlight parasites (UIS4). Module scores on corresponding H&E images show high expression on a scale from low (dark purple) to high (yellow). Scale bars indicate 100 μm. **f** Gene-ontology (GO) enrichment of top five GO-terms of genes associated with proximity (left) or distance to the parasite (right). **g** Loess-smoothed gene expression change (Δ) of host genes exhibiting negative correlation to distance to parasite neighborhoods within 400 μm to parasite neighborhoods across time points of infection. Ribbons around the curve indicate the standard error of the mean (SEM). **h** Change in gene expression (Δ) of host genes that exhibit a positive correlation to parasite neighborhood distances within 400 μm to parasite neighborhoods across time points of infection. Ribbons around the curve indicate the standard error of the mean (SEM).

(Fig. 3b). In contrast to this zonated inflammation, our data suggest a delayed global inflammatory response in SGC-challenged mice compared to *P. berghei* infection. Together with our observation that parasite numbers are increased in intermediate regions of hepatic zonation, this observation suggests that zonated inflammatory response to a high dose-infection may influence parasite survival and assist potential clearance, both in periportal and pericentral areas.

Histological annotations reveal immune cell infiltration resembling focal structures, characterized by increased DNA signal (Fig. 3c). These structures, which we have termed "inflammatory hotspots" (IHSs), follow the same trend as the global inflammatory response, primarily appearing at 12 and 24 hpi in the infected conditions and at a lower frequency at 38 hpi. We explored gene expression profiles correlated with the distance from IHSs and found genes linked to inflammation and immune responses (Supplementary Figs. 15, 16 and Supplementary Data 3). The four genes that exhibit the most obvious decline in expression upon increasing distance to IHSs include *Icam1*, *Gbp2*, *Cxcl9*, and *Cxcl10* (Fig. 3d, e).

Cxcl9 and Cxcl10 are key pro-inflammatory cytokines attracting activated T cells to inflammation sites[46,47]. Gbp2 exhibits antiviral activity in murine macrophages and is upregulated during infection[48]. Icam1 is upregulated by several cell types, including macrophages, and regulates leukocyte recruitment from circulation to inflammation sites[49]. Notably, Cxcl10 upregulation in infected hepatocytes is tied to the previously described abortive parasite phenotype[19]. Additionally, IHSs seem to develop preferentially in periportal proximity (Supplementary Fig. 17).

### snRNA-seq and spatial integration reveal differential expression programs and suggest enrichment of various immune cell types in the IHSs

The addition of snRNA-seq enabled us to define distinct cell populations and their differential gene expression patterns across infection conditions. We deconvolved cell type information and spatial gene expression data to estimate cell type proportions across the tissue.

Comparing the proportions of 14 different annotated cell types (Fig. 4a and Supplementary Data 4), we find 70–80% hepatocytes and 20–30% remaining cell types (Supplementary Fig. 18). Cell type proportions of the 4 identified immune cell clusters (Kupffer cells, monocytes and DCs, T and NK cells and B cells) showed no significant difference in proportions between infected and SGC samples at any time point, but only trends of increased proportions of Kupffer cells, monocytes, and DCs in infected conditions (Fig. 4b).

We explored immune cell expression differences across conditions, noting the upregulation of distinct genes for each immune cell type in infected livers at all time points compared to SGC controls (Fig. 4c, d and Supplementary Data 4). Infection-related marker genes within immune cell types exhibited higher expression at early time points (12 and 24 hpi), declining by 38 hpi. While expression in SGC controls increased over time, it did not reach the same levels as seen in infected cells (Fig. 4c, d).

GO enrichment analysis revealed pathways associated with phagocytosis and leukocyte migration in Kupffer cells (e.g., *Marco*, *Msr1*, *Mertk*, *Cadm1*, *Itga9*, and *Trpm2*). Monocytes and DCs showed upregulation of genes involved in antigen presentation via MHC class II (*H2-Aa*, *H2-Eb1*, *H2-Ab1*, and *Psap*). Lymphoid lineage cells (B, T/NK cells) showed enrichment in leukocyte migration (*Itk*, *Txk*), activation (*Bcl11a*, *Mef2c* for B cells; *Bcl11b*, *Satb1* for T cells), and NK-mediated cytotoxicity (*Cd247*, *Lck*, *Vav3*, and *Prkca*) (Fig. 4e). Thus, GO-term enrichment analysis, along with DGEA, confirms cell types and suggests their heightened activity in *P. berghei* liver infection.

The spatial organization of the different identified cell types across liver tissue sections confirmed the expected anti-correlated distribution of pericentral and periportal hepatocytes across tissue sections (Fig. 5a and Supplementary Fig. 19). This was further validated by proportion-by-distance analysis, using central or portal vein neighborhoods as the center (Fig. 5b).

Pearson correlations between cell type proportions and their distance to parasite neighborhoods across time points identified significant positive or negative correlations (Fig. 5c). Cell types with increased proportions near the parasite included "inflammatory hepatocytes" at 12 and 38 hpi, and pericentral hepatocytes at 24 hpi. Inflammatory hepatocytes are characterized by stress response and inflammation markers (*Saa1*, *Saa2*, *Saa3*, *Ifitm3*, and *Ly6e*) as well as a hepatocyte gene signature (*Alb*, *Apoc3*, *Apoh*, *Hamp*, and *Cyp1e2*). Conversely, cell types with decreased proportions near the parasite included B cells at 12 and 38 hpi and periportal hepatocytes at 24 hpi (Fig. 5c). Despite significant correlation, observed changes in cell type proportions relative to parasite neighborhood distance are small. This suggests that parasites may either have a minor impact on these cell type compositions in the liver tissue, or that only a few cells of these cell types are responsible for the observed differences.

Lastly, we established Pearson correlations between cell type proportions and distances to IHS neighborhoods, jointly analyzing all time points due to the limited number of IHSs. Positive correlations were observed for pericentral hepatocytes in all infection conditions, while negative correlations were observed for cholangiocytes at 38 hpi and in controls. This indicates a preference for IHSs to locate far from pericentral veins and closer to periportal areas (Fig. 5d). Additionally, we noted higher proportions of T/NK cells and monocytes/DCs at IHSs in early infected (12 hpi) sections and 38 h SGC sections (Fig. 5d). These cell types play critical roles in the immune response, as they produce various cytokines and communicate through cytolytic mechanisms[50]. To characterize these cell infiltrates further, we employed IF staining, revealing increased lymphocytic (CD4+, CD8+) and myeloid cell (CD11b+) infiltration and activation over time in infected livers, which also occurs but is delayed in SGC-treated mice (Fig. 5e, f and Supplementary Figs. 20–22, Source data). F4/80+ macrophages within IHSs exhibited the highest abundance at 24 h in infected livers and 38 h in

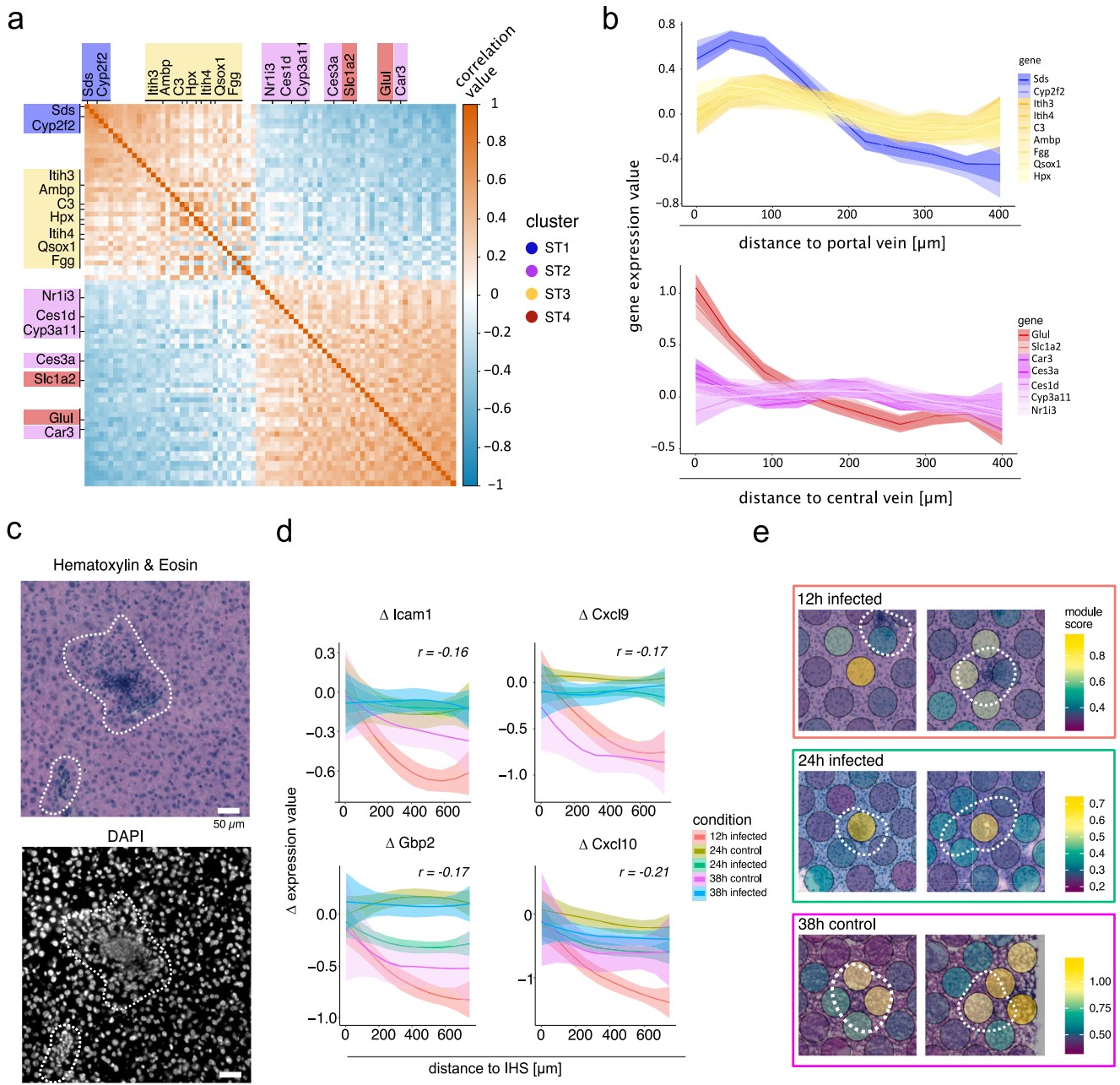

**Fig. 3 | Spatial inflammation in *P. berghei* infected and SGC sections. a** Pearson correlations between marker genes of spots belonging to periportal in cluster ST1 (blue), acute inflammation in cluster ST3 (yellow), pericentral in cluster ST4 (red), and acute pericentral in cluster ST2 (purple). Positive correlation values are indicated in orange and negative correlation values are indicated in blue. **b** Gene expression of genes highlighted in (**a**) as a function of the distance (Loess-smoothed) to the portal vein for marker genes of cluster ST1 and ST3 (top) or the central vein for marker genes of cluster ST2 and ST4. Ribbons around the curve indicate the standard error of the mean (SEM). **c** Representative H&E (top) and DAPI (bottom) images of Inflammatory hotspots (IHSs) observed in *P. berghei* infected section 12 hpi (*n* = 10). IHSs are highlighted with white dotted lines. The scale bar indicates 50 μm. **d** Change in expression (Δ) of the top four genes with the highest

negative correlation as a function of the distance (Loess-smoothed) between 0 and 600 μm from IHSs neighborhoods (Methods for details) where IHSs were present (12, 24, and 38 hpi as well as 24 and 38 h after salivary gland challenge (control)). Ribbons around the curve indicate the standard error of the mean (SEM). **e** Projection of expression modules of genes in (**d**) on tissue sections across three conditions with the highest numbers of visually annotated IHSs (12 and 24 hpi as well as 38 h after salivary gland challenge (control)). Module scores are shown as a color gradient from low scores (dark purple) to high scores (yellow). IHSs are highlighted with white dotted lines. View fields measure 500 by 500 μm. The number of sections per time point and the average number of parasites for each section is indicated at the bottom of each time point.

SGC livers (Fig. 5e, f and Supplementary Figs. 20–22, Source data). Notably, CD27 was exclusively detectable in *P. berghei*-infected livers at all time points, indicating heightened lymphocyte activation compared to controls (Fig. 5e, f and Supplementary Figs. 20–22)[51]. Together with previous studies, where higher proportions of extracellular

matrix-producing mesothelial and mesenchymal cells have been described[52,53] (Fig. 5c), our results suggest that IHSs represent sites of cytolysis or injury followed by tissue regeneration. However, based on the technical limitations, further analyses, beyond the scope of this study, are necessary to validate this hypothesis in greater detail.

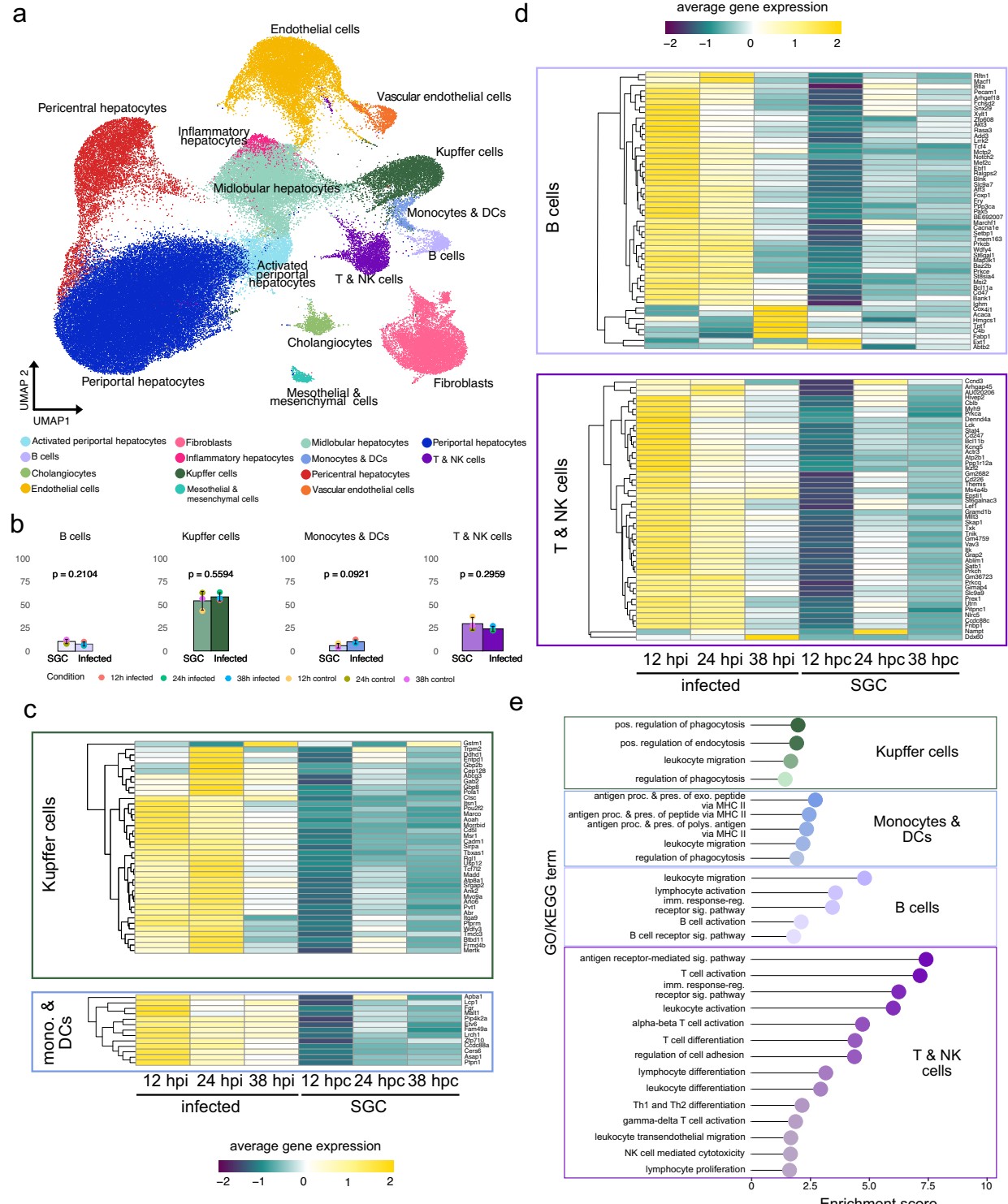

**Fig. 4 | Identification of liver cell types and differential gene expression of immune cell clusters across infection conditions. a** UMAP projection of annotated liver cell types after integration of single-nuclei expression data of all infection conditions: 12, 24, and 38 hpi as well as 12, 24, and 38 h post challenge with salivary gland lysate (SGC). **b** Average immune cell type proportions normalized to the total number of different immune cells (T & NK cells, B cells, monocytes & DCs, and Kupffer cells). The number of samples used to calculate each cell type proportion was $n = 3$ (i.e., one per time point and infection category, as indicated in the legend). An independent two-sample $t$-test was used to compare the mean cell type proportions between infected and control samples, and the resulting $p$ values are shown. Data were presented as mean values ± the standard error of the mean (SEM). **c** Heatmap visualization of differential gene expression of genes associated with

cell types of the myeloid lineage including Kupffer cells, Monocytes (mono.), and dendritic cells (DCs) across infection conditions and time points. Average gene expression across respective cell types is depicted in a color scale ranging from low (purple) to high (yellow). **d** Heatmap visualization of differential gene expression of genes associated with cell types of the lymphatic lineage including B cells, T cells and NK cells across infection conditions and time points. Average gene expression across respective cell types is depicted in a color scale ranging from low (purple) to high (yellow). **e** Gene-ontology (GO) enrichment of GO- or KEGG-terms of unique genes associated with different cell types. Colors indicate the respective immune cell types, including Kupffer cells, monocytes & DCs, B cells, and T & NK cells (top to bottom).

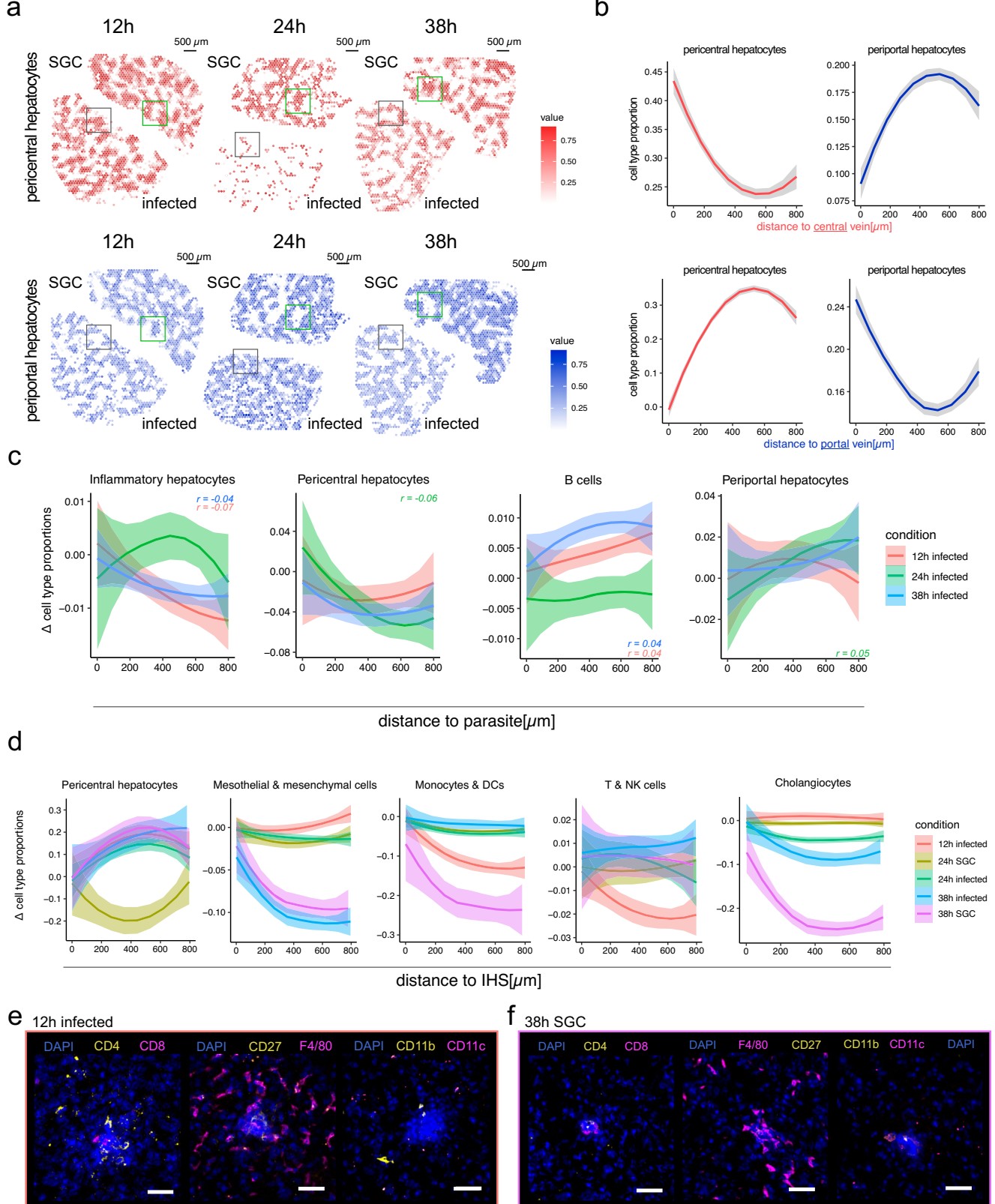

## Discussion

In this study, we employ Spatial Transcriptomics and snRNA-seq to explore host–parasite interactions during *P. berghei* liver stage development in the true tissue context. We uncover spatial elements that impact parasite growth and immune evasion, including tissue-wide and focal inflammatory responses, lipid homeostasis, and liver zonation. Moreover, we evaluate the roles of myeloid and lymphoid immune cells along with other liver resident cells during malaria infection.

Recent advances in next-generation sequencing have greatly enhanced our understanding of multiple stages of the *Plasmodium* life cycle, including liver stage development[54–57]. However, until recently,

**Fig. 5 | Integration of spatial and single-nuclei data. a** Visualization of pericentral (top) and periportal (bottom) cell type proportions across spatial positions of sections generated by 10X Visium protocol. Pericentral cell type proportions are shown in red and periportal cell type proportions in blue. Green and gray boxes highlight smaller regions of opposite cell type compositions in salivary gland lysate control (SGC) and infected sections, respectively. The scale bars indicate 500 μm. **b** Loess-smoothed pericentral and periportal cell type proportions along a distance between 0 and 800 μm originating at computationally annotated central (top) or portal (bottom) veins. Periportal hepatocyte proportions are shown in blue and pericentral cell type proportions in red. Ribbons around the curve indicate the standard error of the mean (SEM). **c** Change in cell type proportions (Δ) of cell types (Loess-smoothed) with significant ($p \leq 0.05$) negative (inflammatory hepatocytes, pericentral hepatocytes) or positive (B cells, periportal hepatocytes) correlation between the distance of 0 to 800 μm to parasite neighborhoods. Ribbons around the curve indicate the standard error of the mean (SEM). Conditions are indicated by colors (12 h infected = red, 24 h infected = green, 38 h infected = blue).

Correlation values ($r$) are indicated for each condition in the respective color. **d** Change in cell type proportions (Δ) of cell types (Loess-smoothed) with significant ($p \leq 0.05$) positive (pericentral hepatocytes) or negative (mesothelial & mesenchymal cells, T & NK cells, and monocytes & DCs) correlation between the distance of 0 and 800 μm to IHS neighborhoods (methods for details) where IHSs were present (12, 24, and 38 hpi as well as 24 and 38 h after salivary gland challenge (control)). Correlations were calculated jointly for all time points and ribbons around the curve indicate the standard error of the mean (SEM). **e** Composite image of immunofluorescent staining of CD4+ and CD8+ cells (left), CD27+, F4/80+ (middle), and CD11b+ and CD11c+ (right) cells in IHSs of tissue sections at 12 hpi ($n = 12$). DNA is stained using DAPI. Individual images and quantification are shown in Supplementary Fig. 20. The scale bars indicate 50 μm. **f** Composite image of immunofluorescent staining of CD4+ and CD8+ cells (left), CD27+, F4/80+ (middle), and CD11b+ and CD11c+ (right) cells in IHSs of SGC tissue sections at 38 h post challenge ($n = 8$). DNA is stained using DAPI. Individual images and quantification are shown in Supplementary Fig. 22. The scale bars indicate 50 μm.

spatial information of host–parasite interactions in liver tissue has been missing. Afriat and colleagues described spatiotemporal interactions at the single-cell level between zonated hepatocytes and *P. berghei* parasites[19]. However, while these studies show transcriptional changes of the parasite and the host in infected hepatocytes as well as expression of selected genes in surrounding cells by smFISH at high resolution, comprehensive investigations within the true tissue context have been lacking. This includes potential paracrine and endocrine interactions of infected hepatocytes and surrounding cells as well as other cell types.

Performing ST in combination with immunofluorescence staining of the intact parasites (UIS4) on the same infected tissue section enabled us to confidently associate transcriptional programs with parasite neighborhoods. We established correlations between gene expression involved in immune and lipid metabolism pathways near parasite neighborhoods at the late stages of infection. Moreover, we showed activation of various immune cell types across the infected liver during infection. Our analyses do not show a correlation of increased immune cell proportion near neighborhoods of successful parasite infection, which suggests that immune cells may be uniformly distributed across the tissue, and their activity may effectively be evaded by successful parasites within the parenchyma.

Lipids are essential for *P. berghei* liver stage development and are scavenged from the host cells by the parasite[58]. At 38 hpi, our data show higher expression of genes involved in lipid metabolism, such as *Fabp5*, close to parasite locations. Fabp5 is known to selectively enhance the activities of PPARß/∂ and PPARγ[37]. PPARs can reduce inflammation by exhibiting anti-inflammatory potential[59–62]. Thus, induced upregulation of expression of *Fabp5* may exhibit a lipid metabolism-dependent evasion strategy induced by the parasite. Meanwhile, *Insig1* expression increases with increased distance from the parasite. The absence of *Insig1* enhances lipid and cholesterol synthesis[35], potentially providing more lipid and cholesterol for the parasite in its proximity.

In combination with the absence of increased immune cell type proportions in proximity to parasite positions at 38 hpi, we speculate that the parasite may achieve a dual positive outcome by changing the lipid composition at the site of infection. While acquiring essential lipids for their development 38 hpi, deregulation of lipid homeostasis may simultaneously assist anti-inflammatory effects by restricting immune cell activity at the site of infection[63–65]. Further, our analyses show upregulation of expression of the autophagy antagonist *Rheb*[40,66] in proximity to parasite locations in the tissues. Downregulation of *Rheb* contributes to ATG7 depletion. ATG7 plays a crucial role in the maturation of autophagosomes[40]. Therefore, the downregulation of *Rheb* may help the parasite to evade autophagy by limiting efficient autophagosome formation.

Upon entering the liver, the parasite crosses the sinusoidal layer and continues to traverse multiple hepatocytes before invading a final hepatocyte, where it initiates replication[67]. The reason for this traversal is still elusive[68], and detailed characterization of the interactions between traversed hepatocytes and immune cell responses remains a subject of investigation. Potentially, IFN-mediated immune responses are triggered by both traversed and infected cells, or result from paracrine crosstalk among infected, traversed, and neighboring immune and parenchymal cells. The high dose of sporozoites in our study may in part explain the global activation of previously reported upregulation of ISGs during infection progression[16–18].

We find that tissue-wide pro-inflammatory responses occur with a delay of 12 to 26 h (at time points 24 and 38 post challenge) in tissue sections from SGC mice. This delayed response is likely triggered by proteins from mosquito salivary glands and residual bacterial material in the saliva. We speculate that this response differs in magnitude from a reaction to a mosquito bite infection, due to intravenous injection of a high number of dissected and lysed salivary glands. Simultaneously, this observation highlights that caution must be taken when interpreting immune responses toward *P. berghei* parasites in experimental set ups that lack salivary gland injection controls. Given the limited coverage area of the Spatial Transcriptomic or Visium slide and the assay's sensitivity, it was crucial to inject a substantial number of parasites to guarantee the detection of multiple parasites within a single section. Upon technological advancement of the spatial omics field, future studies could further explore this finding using lower numbers of parasites, more in line with a natural infection, and comparing it to mosquito bite infections.

Furthermore, we identified inflammatory hotspots (IHSs) with distinct tissue morphology, showing upregulated pro-inflammatory gene signatures nearby. This is supported by increased proportions of various immune cell types and cell surface markers near IHSs. These infiltrates, resembling responses to local inflammation, can have diverse cell compositions and effects on liver health, often involving immune response and regeneration[24,25]. IHSs have been observed in viral diseases like rubella, COVID-19, and Epstein-Barr virus, which affect the liver without causing significant liver disease, usually resulting in subclinical involvement and self-limitation[23,24]. To our knowledge, these focal inflammatory infiltrates, or IHSs, have not previously been reported in the context of malaria. We do not observe co-localization of IHSs with parasites stained with UIS4 antibodies. UIS4 has been ascribed a critical role in avoiding parasite elimination, suggesting that the parasites we detect are still intact[69]. Immune infiltration could be triggered by the parasite's initial traversal through hepatocytes during early invasion, or by parasites that failed to successfully invade or develop early during the liver stage.

In our proposed model, malaria parasites not only resist pro-inflammatory host signals but may actively attenuate inflammation in their vicinity, thereby limiting the infiltration of effector immune cells while simultaneously acquiring essential lipids for their development. This evasion strategy involves the modulation of lipid homeostasis, including PPAR signaling, and the limitation of autophagy. We speculate that the majority of IHSs eliminate parasites early during parasite infection (before 12 hpi), which is why we are unable to observe parasites within fully formed IHSs. Our data also suggests that these structures seem to preferentially form in periportal areas of the liver, supporting the prviously reported, potential importance of liver zonation for parasite survival[19]. Elimination of parasites by IHSs in periportal areas of the tissue would be in line with earlier observations of periportal clearance of pathogens[70]. However, additional studies are necessary to fully characterize their role during malaria development in the liver.

In summary, our study provides a detailed spatiotemporal atlas of the host–parasite interplay during *Plasmodium* development in the liver, in its true tissue context. Malaria eradication efforts require more extensive knowledge of the underlying biology in de novo immunization efforts. To this end, high-resolution, i.e., single-cell spatial omics applications, will be indispensable for understanding the coordination of immune priming in events of partial or full immunization. Future studies that build on the results presented in our study will broaden our understanding of the involvement of lipid metabolism, autophagy, and IHSs during *Plasmodium* infection of the liver, where IHSs will be of particular interest to study in the context of the acquisition of protection against *Plasmodium*.

## Methods

### Ethical statement
The study was performed in strict accordance with the recommendations from the Guide for Care and Use of Laboratory Animals of the National Institutes of Health (NIH). The animal use was done in accordance with the National Institute of Allergy and Infectious Diseases Animal Care and Use Committees (NIAID ACUC), proposal LMVR 22.

### Statistics and reproducibility
No statistical method was used to predetermine the sample size. The number of samples for this study were chosen to include at least two biological replicates for each condition in the ST analysis and one (or more) of the biological replicates of each condition were analyzed using Visium or snRNA-seq data. For ST data, only liver sections that produced data that fulfilled quality control standards were considered. The experiments were not randomized, and investigators were not blinded to allocation during experiments and outcome assessments, apart from the annotation of immunofluorescence signals of parasites and IHSs (ES) and histological structures (NVH) across different liver sections.

### *P. berghei* infections and sample collection
Challenges with *Plasmodium berghei ANKA* (Anka 2.34) sporozoites or salivary gland lysate (*Anopeheles stephensi*) in female 8–9-week-old C57BL/6 mice were performed by tail-vein injection. First, *P. berghei*-infected *A. stephensi* salivary glands were collected 18–21 days post infection and dissected to collect sufficient sporozoites for each challenge. The corresponding number of salivary glands were collected from non-infected mosquitoes for control challenges with salivary gland lysate. Sporozoites and lysate were pelleted by centrifugation, washed, and stored in cold PBS, where the final concentration of sporozoites was determined. Sporozoites were diluted to reach a total number of 300,000–400,000 sporozoites for each infection. After tail-vein injection in the mice, livers were collected after 12, 24, or 38 h. All experimental animals were maintained at

optimal conditions, including 12 h day/night cycles, where mice were housed at ambient temperature and humidity and mosquitoes were maintained at 25–27 °C and 80% humidity.

### Collection and preparation of liver samples
The livers were collected, and the lobes were separated. Each lobe was segmented so cryosections would fit on the 6200 × 6400 µm areas of the Codelink-activated microscope (ST 2k array) or Visium slides and frozen in −30 °C 2-Methylbutane (Merck, cat.no.: M32631-1L). For spatial experiments, the frozen liver samples were embedded in cryomolds (10 × 10 mm, TissueTek) filled with pre-chilled (4 °C) OCT embedding matrix (CellPath, cat.no.: 00411243), frozen and sectioned at 10 µm thickness with a cryostat (Cryostar NX70, Thermo Fisher). Each subarray on the slide is covered with 1934 spots with a 100 µm diameter (ST 2k array), or 4992 spots with a 55 µm diameter (Visium), each containing millions of uniquely barcoded oligonucleotides with poly-$T_{20}$ VN capture regions per spot (barcoded slides were manufactured by 10X Genomics Inc). The full protocol, including sequencing and computational analysis, was performed for a total of 38 (ST) and 8 (Visium) sections. Out of the 38 + 8 samples, 23 (ST), and 4 (Visium) were infected with *P. berghei* parasites. 15 (ST) and 4 (Visium) were challenged with mosquito salivary gland lysate. We analyzed 4 biological replicates for infected samples collected at 12 and 24 h and two biological replicates for infected samples collected at 38 h. For controls, we analyzed livers for 3, 3, and 2 biological replicates, respectively. Samples were selected based on sectioning and RNA quality.

### Immunofluorescence staining of spatial slides
We performed a modified version of the Spatial Transcriptomics workflow according to refs. 71,72. After placing the sections on the ST or Visium slides, they were fixed for 10 min using 4% formaldehyde in PBS (ST) or 30 min in MeOH at −20 °C (Visium). Then, they were dried with isopropanol and parasites were labeled using immunofluorescence as read-out. In short, after fixation, a blocking step using 5% Donkey-serum (Merck, cat.no: D9663-10ML) in PBS for 15 min was performed. Washing steps were performed using a three times concentrated SSC buffer in deionized and RNAse-free water and RNAse Inhibitor (SUPERase•In™ RNase Inhibitor, Thermo Fisher Scientific, cat.no: AM2694), further referred to as blocking buffer. Staining of parasites was performed using an antibody against *Plasmodium berghei* UIS4 produced in goat (Nordic BioSite, cat.no: LS-C204260-400) in a concentration of 1:100 in 1:5 concentrated blocking buffer for 20 min at room temperature. The sections were washed and fluorescently labeled using a Donkey anti-Goat IgG (H + L) Highly Cross-Adsorbed Secondary Antibody, Alexa Fluor Plus 594 (Thermo Fisher Scientific, cat.no: A32758) at a concentration of 1:1000 in 1:5 concentrated blocking buffer for 20 min at room temperature and in the dark. The slides were washed, and DNA was stained using 1:1000 concentrated DAPI solution (Thermo Fisher Scientific, cat.no:62248) for 5 min at room temperature and in the dark. Then, slides were mounted with 85% glycerol (Merck Millipore, cat.no.: 8187091000) including RNAse Inhibitor (SUPERase•In™ RNase Inhibitor, Thermo Fisher Scientific, cat.no: AM2694) and covered with a coverslip. Images were acquired at 20x magnification, using the Zeiss AxioImager 2Z microscope and the Metafer Slide Scanning System (Metasystems).

### Histological staining and annotations
After immunofluorescence staining, a histological staining with Mayer's hematoxylin (Dako, cat.no.: S330930-2) followed by Eosin (Sigma-Aldrich, cat.no.: HT110216-500ML) diluted in Tris/acetic acid (pH 6.0) was performed. The stained sections were mounted with 85% glycerol (Merck Millipore, cat.no.: 8187091000) and covered with a coverslip. Brightfield images were acquired at 20x magnification, using a Zeiss AxioImager 2Z microscope and the Metafer Slide Scanning System

(Metasystems). The liver images were assessed by an expert liver histologist (NVH) who annotated the portal (PV) and central veins (CV), based on the presence of bile ducts and portal vein mesenchyme (PV) or lack thereof (CV). When the quality of the sample did not allow for annotation, "ambiguous vein" was reported. Moreover, regions of apparent cell infiltration (IHSs) were annotated based on increased nuclear signal.

## Permeabilization, cDNA synthesis, tissue removal, and probe release

Next, the slides were put in slide cassettes to enable separated on-array reactions in each chamber, as described previously[71]. Each tissue section was pre-permeabilized using Collagenase I for 20 minutes at 37 °C. Permeabilization was performed using 0.1% pepsin in 0.1 M HCl for 10 min at 37 °C. cDNA synthesis was performed overnight at 42 °C. Tissue removal from the arrays prior to probe release was performed using Proteinase K in PKD buffer at a 1:7 ratio at 56 °C for 1 h. Lastly, the surface probes were released, and cDNA library preparation followed by sequencing was performed.

## ST cDNA library preparation and sequencing

Released mRNA-DNA hybrids were further processed to generate cDNA libraries for sequencing. In short, the second-strand synthesis, cDNA purification, in vitro transcription, amplified RNA purification, adapter ligation, and post-ligation purification, were done using an automated MBS 8000+ system[72]. To determine the number of PCR cycles needed for optimal indexing conditions, a qPCR was performed. After the determination of the optimal cycle number for each sample, the remaining cDNA was indexed, amplified, and purified[73]. The average length of the indexed cDNA libraries was determined with a 2100 Bioanalyzer using the Bioanalyzer High Sensitivity DNA kit (Agilent, cat.no.:5067-4626), and concentrations were measured using a Qubit dsDNA HS Assay Kit (Thermo Fisher, cat.no:Q32851) and libraries were diluted to 4 nM. Paired-end sequencing was performed on the Illumina NextSeq500 (v2.5 flow cell) or NextSeq2000 platform (p2 or p3 flow cell), resulting in the generation of 80 to 150 million raw reads per sample. To assess the quality of the reads FastQC (v 0.11.8) reports were generated for all samples.

## ST spot visualization and image alignment

Staining, visualization, and imaging acquisition of spots printed on the ST slides were performed. Briefly, spots were hybridized with fluorescently labeled probes for staining and subsequently imaged on the Metafer Slide Scanning system (Metasystems). The previously obtained brightfield image of the tissue slides and the fluorescent spot images were then loaded in the web-based ST Spot Detector tool[74]. Using this tool, the images were aligned, and the spots under the tissue were recognized by the built-in recognition tool. Spots under the tissue were then slightly adjusted and extracted.

## Visium experiments

Spatial experiments with increased resolution were carried out using the 10X Visium Spatial Technology (10X Genomics, cat.no: 1000187) according to a slightly modified version of the protocol provided by 10X Visium. In brief, immunofluorescent staining of *P. berghei* parasites using an anti-UIS4 antibody and DNA using DAPI was performed as described above. After fluorescent imaging, Hematoxylin & Eosin (H&E) staining, and brightfield imaging, the tissue was permeabilized for 30 min using the permeabilization buffer provided by the reaction kit. Then, cDNA synthesis, template-switching, and second-strand synthesis were performed according to the manufacturer's instructions. Library generation was performed by amplification and purification of resulting products from the previous steps. Fragment traces were determined with a 2100 Bioanalyzer using the Bioanalyzer High Sensitivity DNA kit (Agilent, cat.no.:5067-4626), and

concentrations were measured using a Qubit dsDNA HS Assay Kit (Thermo Fisher, cat.no: Q32851) and libraries were diluted to 2 nM and pooled for sequencing. Sequencing was performed using a NextSeq2000 (p2 or p3 flow cell) instrument resulting in approximately 80 million reads per sample.

## Single-nuclei RNA sequencing (snRNA-seq)

Nuclei were isolated from snap-frozen liver tissue with a sucrose gradient, as previously described[75]. Briefly, frozen liver tissue was homogenized using the Kimble Dounce grinder set to 1 ml in the homogenization buffer with RNAse inhibitors. Homogenized tissue was then subjected to density gradient (29% cushion – Optiprep) ultracentrifugation (7700 × g, 4 °C, 30 min). Nuclei were resuspended and 2 biological replicates of each condition were pooled before nuclei were stained using DAPI. Intact nuclei were FACS-purified from remaining debris.

A total of 60,000 nuclei were sorted into BSA-coated tubes. The sorted nuclei were pelleted by centrifugation for 3 min at 400 × g and 5 min at 600 × g, sequentially. Nuclei were then resuspended in PBS with 0.04% BSA at ~1000 nuclei/μl. Nuclei suspensions (target recovery of 20000 nuclei) were loaded on a GemCode Single-Cell Instrument (10x Genomics, Pleasanton, CA, USA) to generate single-cell Gel Bead-in-Emulsions (GEMs). Single-cell RNA-Seq libraries were prepared using GemCode Single-Cell 3′Gel Bead and Library Kit (10x Genomics, V2 and V3 technology) according to the manufacturer's instructions. Briefly, GEM-RT was performed in a 96-Deep Well Reaction Module: 55 °C for 45 min, 85 °C for 5 min; end at 4 °C. After RT, GEMs were broken down and the cDNA was cleaned up with DynaBeads MyOne Silane Beads (Thermo Fisher Scientific, 37002D) and SPRIselect Reagent Kit (SPRI; Beckman Coulter; B23318). cDNA was amplified with a 96-Deep Well Reaction Module: 98 °C for 3 mins; cycled 12 times: 98 °C for 15 s, 67 °C for 20 s, and 72 °C for 1 min; 72 °C for 1 min; end at 4 °C. Amplified cDNA product was cleaned up with SPRIselect Reagent Kit prior to enzymatic fragmentation. Indexed sequencing libraries were generated using the reagents in the GemCode Single-Cell 3′ Library Kit with the following intermediates: (1) end repair; (2) A-tailing; (3) adapter ligation; (4) post-ligation SPRIselect cleanup, and (5) sample index PCR. Pre-fragmentation and post-sample index PCR samples were analyzed using the Agilent 2100 Bioanalyzer.

snRNA-seq libraries were pooled in equal ratios and loaded on an S4 lane Illumina NovaSeq 6000, resulting in 2500 – 3000 million read-pairs. Sequencing was performed at the National Genomics Platform (NGI) in Stockholm, Sweden. Spatial (Spatial Transcriptomics, Visium) and snRNA-seq data were aligned to a custom reference genome combining *Mus musculus* (GRCm38.101) and *P. berghei* (PlasmoDB-48_PbergheiANKA) using the stpipeline[76] (v.1.8.1) and STAR (v.2.6.1e) for ST data, spaceranger (v.2.0.0) for Visium data and cellranger (v.3.0.0) for snRNA-seq data.

## Immunofluorescence staining of inflammatory hotspots

We performed IF staining of *P. berghei* infected and control (salivary gland lysate challenged) tissues after 12, 24, and 38 hpi. For each experiment, three consecutive tissue sections of the same tissues utilized for spatial as well as single-nuclei experiments were placed on spatially separated positions of a Super frost slide (VWR, cat.no: 631-0108). After placement, the tissue was fixed using pre-cooled methanol and incubated for 15 min at −20 °C. Tissue sections were permeabilized using 0.2% TritonX-100 (Sigma, cat.no: T8787) in PBS for 5 min and blocked for 15 min using 5% donkey-serum in PBS. After blocking, mouse-specific primary antibodies were applied in different combinations across the three sections. These included (i) 10 μg/ml monoclonal CD4 (Thermo Fisher, cat.no: MA1-146, clone GK1.5), and 10 g/ml monoclonal CD8 (Thermo Fisher Scientific, cat.no: MA5-29682, clone 208), (ii) 1:100 diluted monoclonal F4/80 (Thermo Fisher Scientific, cat.no: MA5-16624, clone CI:A3-1), and 2 μg/ml monoclonal CD27

(Thermo Fisher Scientific, cat.no: MA5-29671, clone 12) and (iii) 10 µg/ml monoclonal CD11b (Thermo Fisher Scientific; cat.no: 53-0112-82, clone M1/70) and 5 µg/ml monoclonal CD11c (Thermo Fisher Scientific, cat.no: 42-0114-82, clone N418). All antibodies were incubated with the tissue for 60 min at room temperature. Tissue sections were washed three times with PBS, and corresponding secondary antibodies (in 1:1000 dilutions) were applied. These included (i) Donkey anti-Rat IgG (H + L) Highly Cross-Adsorbed Secondary Antibody, Alexa FluorTM 488, InvitrogenTM (cat.no: A21208) (ii) Donkey anti-Rabbit IgG (H + L) Highly Cross-Adsorbed Secondary Antibody, Alexa Fluor™ 555 (cat.no: A-31572), (iii) Donkey anti-Rat IgG (H + L) Highly Cross-Adsorbed Secondary Antibody, Alexa FluorTM 647 (cat.no: A78947) and iv) Donkey anti-Rabbit IgG (H + L) Highly Cross- Adsorbed Secondary Antibody, Alexa FluorTM Plus 647 (cat.no: A32795). All antibodies were incubated with the tissue for 30 min at room temperature. Tissue sections were washed three times with PBS, and DNA was stained using 1 µg/ml DAPI (Thermo Fisher Scientific, cat.no: 62248) for 5 min at room temperature. Tissue sections were mounted using Diamond antifade mounting medium (Thermo Fisher Scientific, cat.no: S36972) and imaged. To select inflammatory hotspots which occur in all three consecutive sections, a tiled scan of the DNA counterstain was performed at 20X magnification. Selected hotspots were then imaged at 40X magnification using the same settings across each tissue section. Imaging analysis was performed using ImageJ, where brightness and contrast were adjusted for visualization purposes and composite creation.

## Computational analysis

**Filtering, normalization, integration, dimensionality reduction, and unsupervised clustering.** The main computational analysis of spatial read-count matrices (ST and Visium) was performed using the STUtility package (v 0.1.0)[77] in R (v 4.0.5). The complete R workflow can be assessed and reproduced in the R markdown (see code availability section). Analysis of snRNA-seq data was performed using the Seurat package (v 4.1.1). For ST and Visium data, only protein-coding genes were considered for analysis, and genes of the major urinary protein (Mup) family were filtered due to the large differences in expression between individual mice[19,78]. Gene expression was normalized, accounting for differences in sequencing depth and circadian rhythm due to the differences in dissection time points. Subsequently, normalized expression data was scaled, and highly variable genes were selected using the SCTransform function in Seurat. All samples, biological replicates, and dissection time points were further corrected for batch effects using the harmony package (v.0.1.0)[79]. Thereafter, the first 20 harmony vectors were subjected to Shared Nearest Neighbor (SNN) inspired graph-based clustering via the FindNeighbors and FindClusters functions. For modularity optimization, the Louvain algorithm was used, and clustering was performed at a resolution of 0.35 for clustering granularity.

**Visualization and spatial annotation of clusters.** To visualize the clusters in low-dimensional space for snRNA-seq and spatial data as well as the spot coordinates under the tissue for spatial data, non-linear dimensionality reduction was performed using UMAP. Visualization and annotation of identified clusters in UMAP space (snRNA-seq, ST, Visium) on spot coordinates as well as superimposed on the H&E images (ST, Visium) was performed using the Seurat and STUtility package.

**Differential gene expression analysis and gene modules in space.** To investigate changes in gene expression between selected groups, differential gene expression analysis (DGEA) was performed. Groups for comparison were selected in a supervised (tested conditions) or unsupervised fashion (clustering). Then the FindAllMarkers function of the Seurat package was employed to identify all differentially

expressed genes (DEGs) between all investigated groups, including genes with a logarithmic fold change above 0.25. Only DEGs below an adjusted $p$ value of 0.05 were considered for further downstream analysis. To investigate differentially expressed genes between two groups only, the FindMarkers function of the Seurat package was employed using the same thresholds as described. In both cases, a Wilcoxon rank-sum test was performed to identify differentially expressed genes.

**Functional enrichment analysis.** Functional enrichment of genes of interest was performed using the grpofiler2 package (v.1.0). The algorithm defined in the gost function takes a list of genes and associates them with known functional information sources, establishing statistically significant enriched terms. This package can take data from mouse and several other organisms into account to perform the analysis but lacks data of *P. berghei* or other *Plasmodium* species. Therefore, functional enrichment analysis was only performed for *Mus musculus* genes. We investigated functional enrichment from the KEGG and Gene Ontology (GO) database sources and significance was adjusted using g:SCS (Set Counts and Sizes)[80]. Visualization was performed for the most highly enriched terms and enrichment scores are represented as the negative log10 algorithm of the corrected $p$ value.

**Cluster interaction analysis.** To approximate how expression-based clusters interacted in the tissue space, a simple interaction analysis was carried out as described in detail previously[20]. Briefly, the cluster identity or the four nearest-neighboring spots within a distance threshold were registered, to ensure spots located in the actual physical neighborhood were included in the count, as this assumption might not hold for spots at the edge of the tissue. A binomial test was performed to test for significant over (or under) representation (Cluster interactions) and resulting values were visualized in a heatmap and grouped hierarchically, using complete linkage clustering, in the seaborn package (v.0.12.2) in python (v2.7.18). Since clusters vary considerably in size, a random permutation of cluster positions was performed to investigate which interactions are likely to occur by chance.

**Features as a function of distance.** To investigate the relationship between features of interest (gene expression, proportion values) and the distance to a structure of interest (vasculature, parasites, inflammation hotspots) in the tissue sections, the values of the features of interest were modeled as a function of the distance as previously described[20]. In short, brightfield or fluorescence images were used to create a mask for each structure of interest. As the position of the capture locations relates to the pixel coordinates in the H&E images, the created masks were used to computationally measure the distance from each spot to each selected structure. The distance to a selected structure was defined as the minimal Euclidean distance from the center of each spot to any pixel of the union of all masks.

**Expression-by-distance analysis and distance-based correlation analysis.** After determining distances of spots (capture locations), the distance to each structure of interest was associated with each spot and used for downstream analyses, and visualization was adapted using similar to previously reported visualization approaches[20], and reported in detail in the code provided in the code availability section.

To investigate the relationship between a structure of interest and gene expression in its neighborhood across sections, Pearson correlations between the distance to the structure and expression values of each gene in the spatial gene expression data were performed. Spots within a threshold of 400–800 µm from the region of interest were selected. This was based on the size of the region of interest, with a threshold of 400 µm for smaller structures (e.g., parasites) and a threshold of 800 µm for larger structures (e.g., inflammation

hotspots). After calculating correlations between distance and gene expression values, only adjusted (Bonferroni correction) significant correlations were selected ($p < 0.05$).

Visualization of spatial relationships was carried out by plotting the expression of correlated genes defined as $Y$ over the distance to the structure of interest defined as $X$. To better capture trends of each relationship, Loess smoothing $X \sim Y$ was applied to the data, similar as previously described[20]. To better compare differences between different investigated conditions in some cases, the data were transformed to center around 0 for each condition of interest. This was performed by subtracting the fitted value of the loess regression at the minimal distance from each value in the expression data, maintaining the difference in expression $\Delta Y$ along the distance axis $X$. The ribbons around the smoothed curve represent the standard error of the mean (SEM) as given by the loess algorithm.

**Expression-based classification.** Expression-based classification was performed for central and portal veins as previously described[20] using the hepaquery package (v.0.1). In brief, neighborhood expression profiles were created as described above (features as a function of distance) setting a threshold of 142 pixels, which refers to 400 μm and represents the longest distance between adjacent spot-centers in the same row on an ST slide. After the formation of the neighborhoods, their associated weighted profiles for each gene were assembled. For each neighborhood, expression profile class label predictions were performed employing a logistic regression using the LogisticRegression class from sklearn's (v 0.23.1) linear_model module in python. A l2 penalty was used (regularization strength 1), the number of max iterations was set to 1000, and default values were used for all other parameters. Performance validations were carried out using multiple levels of cross-validation as previously described[20]. To prevent overfitting in the applied model due to the limited number of structures, a reduced set of genes was used for the classification[20].

**Single-cell analysis and cell type annotation.** The raw sequencing data files (.bcl files) were demultiplexed into FASTQ files using cellranger mkfastq (CellRanger v3.1.0, 10x Genomics) with default parameters. The demultiplexed reads were aligned to a custom genome of reference using the CellRanger (10x Genomics) pipeline. The genome of reference was created by combining the genomes of *Mus musculus* (GRCm38.101) and *Plasmodium berghei* (PlasmoDB-48_PbergheiANKA). This resulted in an expression matrix for each of the six sequenced samples (12, 24, and 38 h infected and salivary gland control liver samples), which were individually analyzed. The quality control and clustering steps were performed using the Seurat package (v.4.3.0) and following the standard workflow. The quality control pipeline involved (i) removing genes that were detected in fewer than 10 cells, (ii) filtering out cells with less than 200 genes and more than 5000 genes, (iii) excluding cells with over 15% mitochondrial transcripts, and (iv) discarding all mitochondrial and ribosomal genes from the expression matrix.

Doublets in the data were removed using DoubletFinder (v.2.0.3) with a pk of 0.005, 0.22, 0.24, 0.28, or 0.3, depending on the sample. Following this, the data was normalized and scaled using SCTransform (v.0.3.5) with default parameters. The high variable genes needed to perform a principal component analysis (PCA) were identified using the FindVariableFeatures with the "vst" method.

After initial filtering and doublet removal, gene expression of all investigated conditions was integrated using the Harmony package (v.0.1.0), defining the sample origin as a grouping variable. The FindNeighbors and FindClusters functions were used for clustering, and the Louvain algorithm was employed to cluster the cells with a resolution of 0.3 granularity.

Subsequently, cell type annotations were performed on the integrated data using a twofold strategy. First, an automatic cell type prediction was performed using scmap (v.1.16.0). For this, the top 500 most informative features for annotation were calculated using the selectFeatures function and the steady-state annotated mouse liver dataset "Mouse StSt", generated by ref. 81. Then, the scmap cell pipeline was used to project the cell-type labels from the reference dataset onto our data. Following automatic annotation, a manual annotation step based on canonical marker genes was carried out which involved confirming and refining the obtained results from the automated annotation.

To calculate cell type proportions of immune cells (T and NK cells, B cells, monocytes, DCs, and Kupffer cells) across conditions, the annotated cell data for infected samples (12, 24, and 38 hpi) and for control samples (12, 24, and 38 SGC) were each analyzed across infection time points. The average number of cell types of interest for the infected or control groups were calculated, and their proportions were obtained by dividing the cell type count by the total number of cells of the infected samples or the control samples, respectively. To assess the significance of differences between the three infected (12, 24, and 38 hpi) and the three control samples (12, 24, and 38 SGC), a two-sample t-test was performed using base R (v.4.2.2).

**Single-cell data integration (*stereoscope*).** We integrated our annotated snRNA-seq data using stereoscope (v.0.3.1), a probabilistic method designed for spatial mapping of cell types[82]. In short, stereoscope models both single cell and spatial data as negative binomial distributed, learns the cell type specific parameters and deconvolves the gene expression in each spot into proportion values associated with the respective cell type.

Stereoscope was run with 50,000 epochs and a batch size of 2048 for both sn and st modalities using subset snRNA-seq data and a list of highly variable genes. The annotated snRNA-seq expression matrix was subset to include a minimum of 25 and a maximum of 250 cells per cell type, which were selected randomly. The list of highly variable genes was extracted from the snRNA-seq data using Seurat (v.4.3.0) by first normalizing the data (NormalizeData, default parameters) and then identifying the highly variable genes (FindVariableFeatures, selection.method = "vst", features = 5000).

**Reporting summary**
Further information on research design is available in the Nature Portfolio Reporting Summary linked to this article.

## Data availability
Data is available on GeneExpression Omnibus under accession numbers: GSE268018 (ST data), GSE268068 (Visium data), and (snRNA-seq data) GSE268112. Processed data were also deposited in a Zenodo repository (https://doi.org/10.5281/zenodo.8386528). Source data are provided with this paper.

## Code availability
The code to reproduce the analysis is available on Github (https://doi.org/10.5281/zenodo.12687625)[83]. Instructions for the installation and the workflow of the hepaquery package are available at https://github.com/almaan/ST-mLiver.

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

## Acknowledgements

Figure 1a is adapted from "Spatial transcriptomics to define transcriptional patterns of zonation and structural components in the mouse liver", published in Nature Communications, 2021, and licensed under CC BY 4.0[20]. We would like to thank the National Genomics Infrastructure in Stockholm, funded by SciLifeLab, the Knut and Alice Wallenberg Foundation, and the Swedish Research Council, as well as SNIC/Uppsala Multidisciplinary Center for Advanced Computational Science for assistance with massively parallel sequencing and access to the UPPMAX computational infrastructure. Parts of the computations were performed using resources provided by SNIC through the Uppsala Multidisciplinary Center for Advanced Computational Science (UPPMAX). We thank Christian Gnann for the feedback on the design of the antibody-based tissue staining. We thank Amparo Roig Adam for feedback on the automated annotation of cell types. This study was generously funded by grants from: the Swedish Society for Medical Research (SSMF Establishment Grant), STINT, the Jeansson Foundation, and the Swedish Research Council (VR 2021-05057) to J.A.; the Sven and Lily Lawski Foundation to F.H.; the NIH Distinguished Scholars Program and the Intramural Research Program of the Division of Intramural Research (AI001250-01), National Institute of Allergy and Infectious Diseases (NIAID), NIH to J.V.-R.; Karolinska Institutet (2-195/2021) and the Swedish Research Council (VR 2019-01350) to E.R.A. and N.V.H.; Marie Skłodowska-Curie Individual Fellowship (101027317, "MACtivate") to C.Z.

## Author contributions

J.A., J.L., and J.V.-R. conceived and supervised the study. F.H. performed spatial transcriptomics experiments and analyzed the data. F.H., M.U.I., C.Z., and B.V. extracted single-nuclei and generated libraries. F.H. and M.U.I. analyzed snRNA-seq data. J.V.-R., F.H., and T.P. infected mice and dissected livers. J.V.-R. and T.P. dissected mosquitos. N.V.H. performed histological annotations of liver tissues. E.S. and F.H. researched genes for gene set selections in expression-by-distance to the parasite analysis. S.S. and M.H. performed the stereoscope deconvolution, and F.H. analyzed the data. E.S. and J.Q. performed immunofluorescence staining

and analysis of inflammatory hotspots E.R.A., C.L.S., J.L., and J.V.-R. provided resources for the study. F.H., M.U.I., J.Q., and J.A. wrote the manuscript. All authors provided feedback and edited the manuscript.

## Funding

## Competing interests
S.S., M.H., and J.L. are scientific advisors to 10x Genomics Inc. The remaining authors declare no competing interests.
