## [Peer Review File · Nature Communications]

Host-pathogen interactions in the *Plasmodium*-infected mouse liver at spatial and single-cell resolutionREVIEWER COMMENTS

Reviewer #1 (Remarks to the Author):

this study used two novel approaches including spatial transcriptomics and snRNA-seq to investigate *P. berghei*-infected liver tissues, found gene expression change related to lipid metabolism in response to Plasmodium infection in the proximity of infected hepatocytes, and discovered the presence of inflammatory hotspots with distinct cell type compositions and differential liver inflammation programs along the lobular axis in the malaria-infected tissues, provides informative insights regarding malaria-host interaction to malaria research community.

It would be great for this study to provide more information about part below:

The study used spatial transcriptomics in combination with snRNA-seq, but there is less information about their comparison, how many common cell types identified from two methods, how about gene expression in common cell types. Such as, four immune cell clusters (Kupffer cells, monocytes and DCs, T and NK cells and B cells) were identified in snRNA-seq but not in spatial transcriptomics.

In the spatial transcriptome result (fig1), a lots of ST3 cells in infected tissues but not in control tissue (12h), which cluster in snRNA-seq match ST3? How it happened that a big amount of cells appear or disappear in 12h? you should discuss this.

Reviewer #2 (Remarks to the Author):

Hildebrandt et al. utilized a combined approach of spatial transcriptomics and single nuclear RNA sequencing data to analyze Plasmodium berghei infection of mouse livers. They examined various time points and identified both global and local effects in infected livers. An important control involved injecting non-infected salivary glands, which also induced global changes in the respective mice, albeit with a much delayed onset compared to infected mice. This dataset provides valuable insights into the events occurring at the site of infection, which extend beyond the infected cells to include the surrounding cells, including infiltrated immune cells.

Technical limitations of the study:

- One notable technical limitation of the study is the high number of intravenously injected sporozoites, which likely contributed to the systemic response they observed. This is especially evident as the uninfected salivary gland extracts also triggered an inflammatory response, albeit with a delayed onset of several hours. This delayed response may be attributed to the route of injection (intravenous), which is commonly employed but not physiologically representative. Therefore, it is probable that these general pro-inflammatory responses do not accurately reflect the immune reactions elicited by mosquito bites.

- While the authors achieved high resolution by elegantly combining spatial transcriptomics and single nuclear RNA sequencing, it is currently not possible to distinguish infected cells from surrounding cells.

Overall, the study has been meticulously conducted, and I have no major concerns. However, the authors should consider modifying the text for enhanced clarity. Additionally, they should provide a more detailed discussion of the technical limitations as pointed out above.

It would also benefit the reader if the authors conducted a more comprehensive comparison between their results and the findings of the Afriat study.

The authors should exercise caution in their speculation regarding lipid metabolism as an anti-inflammatory response, as there could be various other reasons for this observation. One of them is, as they correctly pointed out, supplying lipids for the rapidly growing parasite is essential.

Minor concerns

Line 111-115: $38 + 18 = 56$ not 46. Please correct.

Line 112: Replace "infected" with "injected," as the controls involved uninfected salivary glands.

Lines 359-362: Please clarify. They identify an upregulation of Rheb expression near the parasite and suggest that downregulation might support parasite survival. This needs further explanation to establish coherence.

Lines 389-390: this paper refers to liver injury during blood stage, not during liver stage. Please clarify or omit this statement.

Figures: please be more consistent in the use of colors for the ST clusters (fig 1B and D)

Reviewer #3 (Remarks to the Author):

This is an interesting study that evaluated the liver single cell and spatial transcriptome (ST) in response to plasmodium infection. Both ST and single cell data would allow for unbiased transcriptome profiling. Even though the study is potentially impactful, there are several critical points that would need to be addressed before the work could be suitable for publication.

Somehow, the ST data seems to be in a very low resolution, and a single spot seems to contain a lot of cells from diverse origin. This is especially problematic around the infected area and histological lesions (which is termed as inflammation hotspot, IHS). The authors used statistical method to get the pattern of gene expression around lesions and cell recruitment to the plasmodium and IHS, which is impressive. Still, the visualization in relation to infected area is not clear and highly obscured. The relationship between infection, inflammation and metabolic alteration, is obscure. For readers, this could be troubling. More substantiation on claimed findings would be necessary to make this an impactful contribution to the field.

For instance, authors performed analysis about how many different cells are located around plasmodium infected area and IHS (Fig. 5C/D), such as different hepatocytes and immune cells. These were presented in the main figure. The authors did perform immunostaining to substantiate some of these findings, but the results were buried in supplemental data, and it was not compared with the findings from ST. It is suggested that the authors bring those to the main figure, and perform direct side-by-side comparison if their patterns are consistent with what was obtained from ST.

Fig 5A -- Here the authors would need to indicate where are parasite neighborhood, and where are IHS neighborhood. Generally, the concern is that the readers did not clearly indicate the frequency of such structures here. Without such info, it is difficult to judge or evaluate how the authors' analysis is meaningful. Also, even though there is a huge impact of infection on metabolic zonation, it is unclear how infection of IHS spatially specifically produced such impact spatially. From the figure, it seems that the impact is rather widespread over the whole tissue area, not restricted to a specific area. It is conceivable that this is not spatial effect, but a systemic effect, like general hepatitis. If parasite/IHS is indeed widespread, the data interpretation and visualization would be highly challenging, and some of the conclusions need to be re-written.

Fig 3D -- This would need to be substantiated by single molecular RNA in situ hybridization along with parasite transcriptome or staining. Also, what is the relationship between IHS and parasite neighborhood? Again, the readers would want to see the spatial relationship between parasite, IHS, transcriptome differences, and cell recruitment, but they were not nicely visualized. The effect seems to be rather more widespread and uniform.

In summary, the effect of Plasmodium infection on liver transcriptome profile and metabolic zonation seems to be well presented in the paper. But the actual presence of parasite, its relationship with inflammatory hotspot, and the spatial structure affecting liver transcriptome is

less clearly demonstrated. General upregulation of immune component in inflammatory lesion is somewhat expected, so more detailed elaboration of molecular events underlying parasite effect would strengthen the impact of the current work.

REVIEWER RESPONSES

-

“Host-Pathogen Interactions in the Plasmodium-Infected Mouse Liver at Spatial and Single-Cell Resolution”

Reviewer 1:

this study used two novel approaches including spatial transcriptomics and snRNA-seq to investigate *P. berghei*-infected liver tissues, found gene expression change related to lipid metabolism in response to Plasmodium infection in the proximity of infected hepatocytes, and discovered the presence of inflammatory hotspots with distinct cell type compositions and differential liver inflammation programs along the lobular axis in the malaria-infected tissues, provides informative insights regarding malaria-host interaction to malaria research community.

It would be great for this study to provide more information about part below:

- 1) The study used spatial transcriptomics in combination with snRNA-seq, but there is less information about their comparison, how many common cell types identified from two methods, how about gene expression in common cell types. Such as, four immune cell clusters (Kupffer cells, monocytes and DCs, T and NK cells and B cells) were identified in snRNA-seq but not in spatial transcriptomics.

We understand the reviewer’s concern about the missing comparison between the identified cell types in the snSeq and Spatial Transcriptomics data of our study. To clarify the limitations of our study, the spatial transcriptomics technology employed in our study does not have single cell resolution, each barcoded spot is 55µm (10X Visium) or 100µm (Spatial Transcriptomics 2K array) and therefore consists of a small number of cells (1). Therefore, it is unfortunately impossible to define individual cell types in our spatial transcriptomics data. To gain information about the distribution of cell types within the tissue we can however leverage computational algorithms to estimate cell type proportions within each spot from referenced single-cell data. In our study we generated snRNA-seq data originating from the same tissues that were used in our spatial transcriptomics experiments. Only the snRNA-seq data generated in our study has the resolution to inform about the cell type composition within the analyzed tissue. The spatial transcriptomics data can not discern cell types but can inform about the proportions of cell types identified from a single-cell experiment across a tissue section based on a probabilistic method (stereoscope) that uses single cell data to deconvolve the cell mixtures in spatial data (2,3).

To clarify this distinction more prominently and avoid any confusion in our study we have revised the manuscript accordingly:

“Each tissue region that forms a data point in the ST analysis consists of a small mixture of cells. Therefore, we performed additional single-nuclei RNA-sequencing (snRNA-seq) to deconvolve spatial data and artificially increase the resolution in our analyses (Figure 1a). ” (line 124- 127)

Review Figure 1 | Comparison of spatial and snSeq-data. a) Fraction of cell type proportions across spatial clusters. As each spatial position consists of a mixture of cells, no spatial cluster can be assigned a single cell type. Cell types are shown as different colors and clusters are listed on the x-axis. Spatial clusters showing upregulation of periportal (ST1), pericentral (ST2, ST4) or small spatial clusters exhibiting pro-inflammatory signals (ST10) show higher proportions of periportal hepatocytes, pericentral hepatocytes or Monocytes & DCs, respectively. **b)** Violin plots showing expression levels of three ST3 markers (Saa1, Marco & Itih3) across identified cell types in snSeq data, highlighting that expression of these markers is not exclusive to one cell type.

- (1) <https://pubmed.ncbi.nlm.nih.gov/34857782/>
- (2) <https://pubmed.ncbi.nlm.nih.gov/33037292/>
- (3) https://docs.scvi-tools.org/en/stable/user_guide/models/stereoscope.html

In the spatial transcriptome result (fig1), a lots of ST3 cells in infected tissues but not in control tissue (12h), which cluster in snRNA-seq match ST3? How it happened that a big amount of cells appear or disappear in 12h? you should discuss this.

In light of the concern of comment 1) of the reviewer, we would like to begin addressing this concern by reiterating that each measurement (spot) within the spatial data (ST1-ST12 and V1-V10) correspond to a small mixture of cells and do not represent single cells. Therefore, we can not make a direct correlation between clusters/cell type annotations of the snSeq and clusters identified in the spatial data within our study.

The differentiation between cells and spots is also important to clarify in order to address the “appearance” or “disappearance” of **spots** of cluster ST3 in infected livers compared to control livers at 12 hours post injection of infectious sporozoites or mosquito salivary glands, respectively. We disagree with the phrasing “appearance” or “disappearance” of spots belonging to a cluster and also would like to note that we do not use this phrasing in our manuscript in this context. To clarify with regards to the different numbers of spots associated with a gene expression profile of cluster ST3 in infected liver sections 12 hours post infection compared to the control liver sections at the same time point, we consider it important to recapitulate that gene expression profiles define the annotation of spots to distinct clusters. To visualize the different clusters, which are based on gene expression, we projected these clusters on the dimensionality reduction graph (UMAP), splitting the spatial positions by time point and infection in Figure 1.

As described in the manuscript (line 157-161) and shown in Supp. Figure 5 and Supp. Data 1, cluster ST3 is characterized by expression of genes involved in stress response and the onset of an immune response. The higher number of spots with this expression profile at infected conditions at 12 hours when compared to controls indicates that these responses are only upregulated in infected conditions and not in controls at this time point as discussed in detail in our manuscript (line 137-139). To address any potential confusion and to clarify our findings, we made several adjustments to the manuscript between lines 119 and 177.

We hope we were able to clarify and address the reviewer’s concerns in enough depth to relieve them from any doubt.

Reviewer 2:

Hildebrandt et al. utilized a combined approach of spatial transcriptomics and single nuclear RNA sequencing data to analyze *Plasmodium berghei* infection of mouse livers. They examined various time points and identified both global and local effects in infected livers. An important control involved injecting non-infected salivary glands, which also induced global changes in the respective mice, albeit with a much delayed onset compared to infected mice. This dataset provides valuable insights into the events occurring at the site of infection, which extend beyond the infected cells to include the surrounding cells, including infiltrated immune cells.

Technical limitations of the study:

- 1) One notable technical limitation of the study is the high number of intravenously injected sporozoites, which likely contributed to the systemic response they observed. This is especially evident as the uninfected salivary gland extracts also triggered an inflammatory response, albeit with a delayed onset of several hours. This delayed response may be attributed to the route of injection (intravenous), which is commonly employed but not physiologically representative. Therefore, it is probable that these general pro-inflammatory responses do not accurately reflect the immune reactions elicited by mosquito bites.

We thank the reviewer for pointing out this important distinction relating to experimental vs natural/mosquito infections. Since this study is the first of its kind, i.e. implementing spatial transcriptomics to characterize host-pathogen interactions in the infected liver, the ambition was to include i) controllable infections, with a known/reproducible number of sporozoites, as well as ii) defined volume of mosquito salivary gland factors and iii) specifically, for this first of its kind study, provide an ample number of infected cells within each tissue section in order to enable statistically relevant measurements. We anticipate that this study will set the baseline for the implementation of spatial omics tools to study host-pathogen interactions in tissues, and as such, inform future follow-up studies in the field of infection biology.

In line with the reviewer's comment, tail-vein infections are commonly employed within the field to study sporozoite invasion of liver tissue and parasite development in hepatocytes, and we believe that our study, due to its unprecedented spatial resolution, provides critical information of host-responses due to experimental infections.

The decision to utilize a large quantity of sporozoites stemmed from the inherent challenges in locating infected hepatocytes within liver tissue. Based on our experience in microscopy-based detection of infected liver cells, injecting a substantial number of sporozoites (100-400 k) becomes imperative to identify numerous infected cells within a specific region or section. Given the limited coverage area of the Spatial Transcriptomic or Visium slide and the assay's sensitivity, it was crucial to guarantee the detection of multiple parasites within a single section. This approach facilitates a more comprehensive and precise analysis of how various liver zones respond to infection.

Detecting an ample number of sporozoites through natural mosquito bites poses significant challenges due to substantial variability. Mosquitoes typically transmit a reduced quantity of sporozoites (~800) during each bite (1), with fewer than 100 reaching

circulation and even fewer reaching the liver. Thus, to ensure sufficient infected cells detectable by our assay, a single mouse would need to be bitten by thousands of mosquitoes, which will trigger a whole new set of inflammatory reactions due to the overwhelming number of bites. Using the tail-vein injection approach in our study is necessary to compensate for the low number of sporozoites delivered per bite and achieve reliable detection with our experimental setup. We believe that it is well beyond the scope of this study to perform natural infections, including the necessary further optimizations of the spatial transcriptomics workflow, but instead anticipate that such optimizations would be well-suited and highly informative as a follow up study to our current study.

This has been further clarified in the manuscript:

*“We speculate that this response differs in magnitude from a reaction to a mosquito bite infection, due to intravenous injection of a high number of dissected and lysed salivary glands. Simultaneously, this observation highlights that caution must be taken when interpreting immune responses towards *P. berghei* parasites in experimental set ups that lack salivary gland injection controls. Given the limited coverage area of the Spatial Transcriptomic or Visium slide and the assay’s sensitivity, it was crucial to inject a substantial number of parasites to guarantee the detection of multiple parasites within a single section. Upon technological advancement of the spatial omics field, future studies could further explore this finding using lower numbers of parasites, more in line with a natural infection and comparing it to mosquito bite infections.”*

(line 395-403)

- (1) <https://pubmed.ncbi.nlm.nih.gov/38272943/>
- 2) While the authors achieved high resolution by elegantly combining spatial transcriptomics and single nuclear RNA sequencing, it is currently not possible to distinguish infected cells from surrounding cells.

We thank the reviewer for their feedback and agree with them on the technical limitations of our study. We have elaborated further on these limitations as requested by the reviewer in our answer to their comment 4).

- 3) Overall, the study has been meticulously conducted, and I have no major concerns. However, the authors should consider modifying the text for enhanced clarity.

We appreciate the reviewer's comment and gladly modify the text to enhance clarity. While we would appreciate the reviewer pointing out which sections remain unclear, we made modifications that - in our opinion - enhance clarity. In addition to the other modifications implemented to address reviewers' comments, we made several modifications to the description of our results between line 119-177 specifically to address this reviewer's concerns about clarity.

“The data further shows that the immune response in malaria infected tissue has strong spatial components, including inflammation programs that differ depending upon the lobular zone, as well as regions with enrichment of distinct cell types, that we term ‘inflammatory hotspots’.” (line 40-42)

“Inside the hepatocyte, the parasite resides within a parasitophorous vacuole (PV), surrounded by a parasitophorous vacuole membrane (PVM), which is formed by invagination of the host cell membrane assists it in evading detection by the immune system. Moreover, Plasmodium can exploit this interaction surface to alter the hepatocyte to access nutrients and other essentials required for its growth and development.” (line 54-58)

“Each lobule describes a hexagonal unit with portal veins at the corners and a central vein at the center. To ensure optimal metabolic activity and to prevent futile cycles liver cells express differential proteins along the axis from each portal node to the central vein, defining metabolic zones. This spatial organization is commonly referred to as zonation.” (line 68-71)

“We validated our transcriptional analysis by establishing that these IHS are enriched in immune cell infiltrates, leading us to propose that they are a distinct spatial feature of the immune response to *P. berghei* infection.” (line 110-112)

As well as several modifications to clarify the description of our results between line 119-177.

Lines:

- 4) Additionally, they should provide a more detailed discussion of the technical limitations as pointed out above.

To discuss the technical limitations of our study we modified the discussion and added that the general pro-inflammatory responses observed in our analysis do likely not accurately reflect the immune reactions elicited by mosquito bites because we i) injected mosquito salivary gland by intravenous injection and ii) injected salivary gland lysate of 100 to 150 mosquitos across 6 mice (~ 25 mosquitoes/mouse):

“We find that tissue-wide pro-inflammatory responses occur with a delay of 12 to 26 hours (at time points 24 and 38 post-challenge) in tissue sections from SGC mice. This delayed response is likely triggered by proteins from mosquito salivary glands and residual bacterial material in the saliva. This response likely differs from a reaction to a mosquito bite infection, due to intravenous injection of a high number of dissected and lysed salivary

glands. These observations highlight that caution must be taken when interpreting immune responses towards P. berghei parasites in experimental set ups that lack salivary gland injection controls. Future studies could further explore this finding using lower numbers of parasites, more in line with a natural infection and comparing it to mosquito bite infections.”
(line 392-403)

We also discussed the technical limitation of resolution of the spatial transcriptomics technology and highlighted the complementary study of Afriat and colleagues in this context:

“Afriat and colleagues described spatio-temporal interactions at the single-cell-level between zoned hepatocytes and P. berghei parasites. However, while these studies show transcriptional changes of the parasite and the host in infected hepatocytes as well as expression of selected genes in surrounding cells by smFISH at high resolution, comprehensive investigations within the true tissue context have been lacking. This includes potential paracrine and endocrine interactions of infected hepatocytes and surrounding cells as well as other cell types.”
(line 350-355)

- 5) It would also benefit the reader if the authors conducted a more comprehensive comparison between their results and the findings of the Afriat study.

We thank the reviewer for this suggestion and gladly follow the reviewer’s suggestion to compare our findings more comprehensively with the complementary publication by Afriat et al. (1). In order to follow their suggestion we elucidated further on the discrepancies between both studies as well as how our study can overcome limitations of the Afriat et al. study and vice versa. We partially addressed the reviewer’s comment in our answer to the previous reviewer comment 4.

(line 350-355)

- (1) <https://pubmed.ncbi.nlm.nih.gov/36352220/>

- 6) The authors should exercise caution in their speculation regarding lipid metabolism as an anti-inflammatory response, as there could be various other reasons for this observation. One of them is, as they correctly pointed out, supplying lipids for the rapidly growing parasite is essential.

We agree with the reviewer that an anti-inflammatory response is not the only or the most obvious reason for the increased expression of genes involved in lipid metabolism in the vicinity of the parasite. Therefore, we followed the reviewer’s suggestion and have

disclaimed and highlighted the variety of processes that are potentially affected by the deregulation of lipid metabolism in the vicinity of the parasite, listing anti-inflammatory responses as only one of several factors that may assist the parasite during its development in the liver:

“In combination with the absence of increased immune cell type proportions in proximity to parasite positions at 38 hpi, we speculate that the parasite may achieve a dual positive outcome by changing the lipid composition at the site of infection. While acquiring essential lipids for their development and membrane expansion at 38 hpi, deregulation of lipid homeostasis may simultaneously assist anti-inflammatory effects by restricting recruitment of effector cells of the innate immune response to the site of infection.” (line 347-376)

and

“In our proposed model, malaria parasites not only resist pro-inflammatory host signals but may actively attenuate inflammation in their vicinity, thereby limiting the infiltration of effector immune cells while simultaneously acquiring essential lipids for their development. This evasion strategy involves the modulation of lipid homeostasis, including PPAR signaling and limitation of autophagy. We speculate that the majority of IHSs eliminate parasites early during parasite infection (before 12 hpi), which is why we are unable to observe parasites within fully formed IHSs. Our data also suggests that these structures seem to preferentially form in periportal areas of the liver, supporting the potential importance of liver zonation for parasite survival, previously reported by Afriat et al. Elimination of parasites by IHSs in periportal areas of the tissue would be in line with earlier observations of periportal clearance of pathogens. However, additional studies are necessary to fully characterize their role during malaria development in the liver.” (line 417-427)

7) Line 111-115: $38 + 18 = 56$ not 46. Please correct.

The number of sections (46), we mention in our manuscript is correct but we understand the confusion our phrasing created. We analyzed 38 liver sections using the spatial transcriptomics protocol and 8 additional liver sections using the “10X Visium” protocol. The sections analyzed in our analyses were collected from a total of 18 female mice, resulting in multiple replicate sections for each mouse in our analysis. To clarify we changed the respective lines in the revised manuscript to:

*“We used Spatial Transcriptomics (ST) to analyze 38 liver sections infected with either *P. berghei* parasites or injected with *An. gambiae* salivary gland lysate (SGC) at different time points (12, 24, and 38 hours post-infection). We added Visium Spatial Gene Expression analysis of 8 additional liver sections for higher spatial resolution (see Methods*

for details), resulting in a total of 46 spatially analyzed liver sections collected from 18 adult female mice.”

” (line 119-123)

- 8) Line 112: Replace "infected" with "injected," as the controls involved uninfected salivary glands.

We corrected the respective lines in the revised manuscript (line 120).

- 9) Lines 359-362: Please clarify. They identify an upregulation of Rheb expression near the parasite and suggest that downregulation might support parasite survival. This needs further explanation to establish coherence.

We apologize for the confusion and corrected the corresponding section in the text and elaborated further on the proposed hypothesis, which should be tested in future experiments but is beyond the scope of this study:

“Further, our analyses show upregulation of expression of the autophagy antagonist Rheb in proximity to parasite locations in the tissues. Downregulation of Rheb contributes to ATG7 depletion. ATG7 plays a crucial role for the maturation of autophagosomes. Therefore, downregulation of Rheb may help the parasite to evade autophagy by limiting efficient autophagosome formation ” (line 378-382)

- 10) Lines 389-390: this paper refers to liver injury during blood stage, not during liver stage. Please clarify or omit this statement.

We removed this statement from the manuscript to avoid confusion.

- 11) Figures: please be more consistent in the use of colors for the ST clusters (fig 1B and D)

We understand the reviewer’s confusion regarding the colors in figure 1B and D. The colors in Figure 1D are the same but with a changed opacity in an attempt to increase readability of the text. However, we have set the opacity to the same as in figure 1D to avoid any confusion. For ease of inspection we added the updated figure here as Review figure 1.

Review Figure 1 | Updated Figure 1 “Spatial organization of livers infected with *P. berghei* parasites or SGC.” The opacity for figure 1d was set to 100% to appear more consistent with the color scheme in figure 1b.

Reviewer 3:

This is an interesting study that evaluated the liver single cell and spatial transcriptome (ST) in response to plasmodium infection. Both ST and single cell data would allow for unbiased transcriptome profiling. Even though the study is potentially impactful, there are several critical points that would need to be addressed before the work could be suitable for publication.

Somehow, the ST data seems to be in a very low resolution, and a single spot seems to contain a lot of cells from diverse origin. This is especially problematic around the infected area and histological lesions (which is termed as inflammation hotspot, IHS). The authors used statistical method to get the pattern of gene expression around lesions and cell recruitment to the plasmodium and IHS, which is impressive. Still, the visualization in relation to infected area is not clear and highly obscured. The relationship between infection, inflammation and metabolic alteration, is obscure. For readers, this could be troubling. More substantiation on claimed findings would be necessary to make this an impactful contribution to the field.

- 1) For instance, authors performed analysis about how many different cells are located around plasmodium infected area and IHS (Fig. 5C/D), such as different hepatocytes and immune cells. These were presented in the main figure. The authors did perform immunostaining to substantiate some of these findings, but the results were buried in supplemental data, and it was not compared with the findings from ST. It is suggested that the authors bring those to the main figure, and perform direct side-by-side comparison if their patterns are consistent with what was obtained from ST.

We appreciate the reviewer's comment and have moved the immunofluorescence images for the two most prominent conditions (12 hpi and 38 controls) in our data to the main figure to highlight the distribution of immune cells in IHSs. For ease of inspection we added the updated version of Figure 5 here as Review Figure 2. We have also changed the corresponding figure legend and text referring to figure 5 and the supplementary figures, on lines 321-324 and 946-952.

Review Figure 2 | Updated Figure 5 : Integration of spatial and single nuclei data for panel e) Composite image of immunofluorescent staining of CD4+ and CD8+ cells (left), CD27+, F4/80+ (middle) and CD11b+ and CD11c+ (right) cells in IHSs of tissue sections at 12 hpi. DNA is stained using DAPI. Individual images and quantification are shown in Supplementary figure 20). **f)** Composite image of immunofluorescent staining of CD4+ and CD8+ cells (left), CD27+, F4/80+ (middle) and CD11b+ and CD11c+ (right) cells in IHSs of SGC tissue sections at 38h post challenge. DNA is stained using DAPI. Individual images and quantification are shown in Supplementary figure 22). (line 946-952)

2) Fig 5A -- Here the authors would need to indicate where are parasite neighborhood, and where are IHS neighborhood. Generally, the concern is that the readers did not clearly indicate the frequency of such structures here. Without such info, it is difficult to judge or

evaluate how the authors' analysis is meaningful. Also, even though there is a huge impact of infection on metabolic zonation, it is unclear how infection of IHS spatially specifically produced such impact spatially. From the figure, it seems that the impact is rather widespread over the whole tissue area, not restricted to a specific area. It is conceivable that this is not spatial effect, but a systemic effect, like general hepatitis. If parasite/IHS is indeed widespread, the data interpretation and visualization would be highly challenging, and some of the conclusions need to be re-written.

If we understand the reviewer correctly, they would like us to project parasite as well as IHS positions on spot projections of pericentral and periportal hepatocyte proportions across tissue sections in Figure 5a. Figure 5a and Figure 5b show the distribution of cell type proportions of pericentral and periportal hepatocytes which we annotated in the snSeq data. It is unclear to us how information of the position and numbers of IHSs and parasites in Figure 5a would strengthen our observation of general metabolic zonation in Figure 5a and b. This phenomenon is shown previously on the RNA level using spatial transcriptomics (1) and single-cell RNA sequencing (2), as well as on the protein level (3,4).

We assume that the reviewer implies that IHS and/or parasite positions have an impact on zonation. However, we do not observe that these structures change the spatial distribution of pericentral or periportal zones. However, as the reviewer pointed out correctly with their statement “it seems that the impact is rather widespread over the whole tissue area”, we observe a general change in the expression profile in the vicinity of pericentral and periportal zones as shown in Figure 3a and 3b and discussed in our manuscript (line 348-262). Moreover, we describe the relationship between parasites and pericentral and periportal zones in supplementary figure 14 and in our manuscript (line 241-247) as well as the distribution of pericentral and periportal hepatocyte proportions, which align well with spatial gene expression patterns (figure 5a-b, supplementary figure 19 and line 304-307).

In addition, we agree with the reviewer that including the frequency of IHSs and parasites across conditions would be beneficial for the reader to evaluate the impact of our findings. Though, we do believe that these numbers should be included in respect to figure 2 and figure 3, where host gene expression changes in relation to the parasite and IHS positions are discussed, respectively. Therefore, we included these numbers in figure 2d and figure 3e and changed the figure legend accordingly. For ease of inspection of these changes we attach these figures as Review Figure 3 and Review Figure 4. We also include an example of the distribution of parasites and IHSs for each relevant condition as Review Figure 5 here.

Review Figure 3 | edited version of Figure 2 with corresponding changes to the figure legend for panel **d**) Immunofluorescence and Hematoxylin and Eosin (H&E) stained images of *P. berghei* infected tissue sections across investigated conditions at 12h, 24h and 38h (left to right). Colored boxes indicate time points (12 hpi = red, 24 hpi = green, 38 hpi = blue). Positions with parasites are shown from individual IF images, showing DNA staining (DAPI), parasite staining (UIS4) and the composite image (merge). Parasites are highlighted by white circles and scale bars indicate 100 μ m. The position of detected parasites is shown as a black box on the respective H&E images, recorded after immunofluorescent staining. The number of sections per time point and average number of parasites for each section is indicated on the bottom of each timepoint. (line 858-865)

Review Figure 4 | edited version of Figure 3 with corresponding changes to the figure legend for panel **e**) Projection of expression modules of genes in d) on tissue sections across three conditions with highest numbers of visually annotated IHSs (12 and 24 hpi as well as 38h after salivary gland challenge (control)). Module scores are shown as a color gradient from low scores (dark purple) to high scores (yellow). IHSs are highlighted with white dotted lines. View fields measure 500 by 500 μm . The number of sections per time point and average number of parasites for each section is indicated on the bottom of each timepoint. (line 895-897)

Review Figure 5 | Representative Illustration of **a)** parasite annotations and distances to parasite annotations in infected liver sections, **b)** inflammatory hotspots (IHSs) in infected sections and, **c)** IHSs in control sections. Distances to annotated structures are shown as a color gradient from close proximity (dark) to further distance (light) and overlaid on Hematoxylin & Eosin images.

- (1) <https://pubmed.ncbi.nlm.nih.gov/34857782/>
- (2) <https://www.ncbi.nlm.nih.gov/pmc/articles/PMC5321580/>
- (3) <https://pubmed.ncbi.nlm.nih.gov/3062788/>
- (4) <https://pubmed.ncbi.nlm.nih.gov/37783884/>

3) Fig 3D -- This would need to be substantiated by single molecular RNA in situ hybridization along with parasite transcriptome or staining. Also, what is the relationship between IHS and parasite neighborhood? Again, the readers would want to see the spatial relationship between parasite, IHS, transcriptome differences, and cell recruitment, but they were not nicely visualized. The effect seems to be rather more widespread and uniform. In summary, the effect of Plasmodium infection on liver transcriptome profile and metabolic zonation seems to be well presented in the paper. But the actual presence of parasite, its

relationship with inflammatory hotspot, and the spatial structure affecting liver transcriptome is less clearly demonstrated. General upregulation of immune component in inflammatory lesion is somewhat expected, so more detailed elaboration of molecular events underlying parasite effect would strengthen the impact of the current work.

We thank the reviewer for their comment and agree with them that our manuscript would profit from highlighting the link between IHS and parasite neighborhoods as well as further validation of our findings in figure 3D, where we illustrate the top 4 genes which which exhibited a strong negative correlation between expression and distance to IHS neighborhoods.

We extended our UIS4 staining results by staining liver sections using anti-serum from mice infected with *P. berghei* sporozoites to detect a broader range of parasite antigens, and rates of detection of a parasite within an IHS were low (0.03, n.s). Sections of livers 38 SGC also served as a control for negative parasite signal in IHSs.

These results reflect our hypothesis that the IHS are a consequence of immune clearance of the parasite. We speculate that we detect parasites at an IHS only prior to successful elimination of the parasite. I.e. that the probability of detecting parasites within IHSs is highly unlikely, happens only at early stages of IHS development and reflects a snapshot of this dynamic process. In support of this, we note that the IHS where we detect parasites appear to be smaller, reflecting IHS typical of earlier rather than later time points (example image below, Review Figure 6). We agree with the reviewers that characterizing the relationship between parasite detection, IHS formation, and parasite clearance will be a valuable future study, but we believe that it falls outside the scope of this manuscript. However, we adjusted the claims in the manuscript accordingly to phrase the connection between IHS development and parasite infection more speculatively, given the existing relationship of the (i) observation of IHSs at early time points post infection (12hpi) as opposed of their late occurrence in control sections, (ii) the cellular composition of IHSs and (iii) the relationship between IHSs and portal areas across the tissue. We adjusted the manuscript text accordingly and removed Figure 6, which suggests the working model of parasite clearance by IHS formation as we will substantiate this hypothesis in a follow-up study instead.

"In our proposed model, malaria parasites not only resist pro-inflammatory host signals but may actively attenuate inflammation in their vicinity, thereby limiting the infiltration of effector immune cells while simultaneously acquiring essential lipids for their development. This evasion strategy involves the modulation of lipid homeostasis, including PPAR signaling and limitation of autophagy. We speculate that the majority of IHSs eliminate parasites early during parasite infection (before 12 hpi), which is why we are unable to observe parasites within fully formed IHSs. Our data also suggests that these structures seem to preferentially form in periportal areas of the liver, supporting the potential importance of liver zonation for parasite survival, previously reported by Afriat et al. Elimination of parasites by IHSs in periportal areas of the tissue would be in line with

earlier observations of periportal clearance of pathogens. However, additional studies are necessary to fully characterize their role during malaria development in the liver.” (line 417-427)

Review Figure 6 | Representative image of an inflammatory hotspot (IHS) (far left, DAPI) where a parasite is detected by immunostaining with *P. berghei* antiserum (left) and UIS4 antibody (right), within the IHS (far left, combined image).

To address the second part of the reviewer’s comment, we wanted to substantiate our findings on gene-expression gradients around IHSs neighborhoods further. We appreciate the reviewer’s suggestion to attempt single molecular RNA in situ hybridization along with parasite transcriptome or staining. However, the samples generated for this study were specifically processed to perform ST analysis and not processed according to the requirements for high quality single molecular RNA in situ hybridization. Therefore, we validated our findings by immunohistochemistry.

To further validate the results shown in Figure 3D, we stained the intracellular Icam1 and Gbp2 proteins, as Cxcl9 and Cxcl10 encode chemokines, which are secreted by activated cells and highly dispersed across the tissue. Our results clearly show enrichment of both ICAM-1 and GBP2 at IHS neighborhoods in 12 hpi liver sections. In contrast, at 38hpi, IHS are smaller, rarely show enrichment of ICAM-1, and have only weak enrichment of GBP2. This reflects the change in correlation of expression with distance from IHS observed for these genes in our analysis (Review Figure 7).

Review Figure 7 | Immunohistochemistry images of *P. berghei* infected liver tissue sections at 12 hpi (red coloured box) and 38 hpi (blue coloured box). IHS are highlighted by white circles (100µm diameter), in consecutive sections stained for either ICAM-1 and hematoxylin or GBP2 and hematoxylin.

REVIEWER COMMENTS

Reviewer #1 (Remarks to the Author):

The authors addressed the reviewer's concerns and revised the manuscript according to reviews' comments, so it is ready to be accepted now.

Reviewer #2 (Remarks to the Author):

The authors have addressed all my concerns convincingly.

Reviewer #3 (Remarks to the Author):

It's unfortunate but interesting to see that snRNA-seq and ST observations are not aligning well and rather show different aspects. This is clearly a limitation which should be further explicitly mentioned, but at the same time interesting difference. It's acceptable that they monitor different aspects, but since the tissue is the same and gene expression is the same, the observations should be at least consistent with each other. In the reviewer figure, the authors showed the level of Saa1/Marco/Itih3 genes. However, this was only shown for a single set, which is not informative. Could authors stratify the cells according to the conditions and compare the level of these expressions between control/infected at different time point? Could the authors make sense of such data by correlating this data with abundances of different ST clusters in each sample? Whether the ST differences are caused by different cell type abundance or gene expression changes is an important question.

Response to reviewer 3

“Host-Pathogen Interactions in the Plasmodium-Infected Mouse Liver at Spatial and Single-Cell Resolution”

Reviewer 3: It's unfortunate but interesting to see that snRNA-seq and ST observations are not aligning well and rather show different aspects. This is clearly a limitation which should be further explicitly mentioned, but at the same time interesting difference. It's acceptable that they monitor different aspects, but since the tissue is the same and gene expression is the same, the observations should be at least consistent with each other. In the reviewer figure, the authors showed the level of Saa1/Marco/Ith3 genes. However, this was only shown for a single set, which is not informative. Could authors stratify the cells according to the conditions and compare the level of these expressions between control/infected at different time point? Could the authors make sense of such data by correlating this data with abundances of different ST clusters in each sample? Whether the ST differences are caused by different cell type abundance or gene expression changes is an important question.

The reviewer reiterates an interesting and important question previously raised by reviewer 1, whether differences in spatial gene expression observed in our study result from different cell type abundance or gene expression of the same cell types.

See previous response to reviewer 1 below (highlighted in gray):

“We understand the reviewer’s concern about the missing comparison between the identified cell types in the snSeq and Spatial Transcriptomics data of our study. To clarify the limitations of our study, the spatial transcriptomics technology employed in our study does not have single cell resolution, each barcoded spot is 55µm (10X Visium) or 100µm (Spatial Transcriptomics 2K array) and therefore consists of a small number of cells (1). Therefore, it is unfortunately impossible to define individual cell types in our spatial transcriptomics data. To gain information about the distribution of cell types within the tissue we can however leverage computational algorithms to estimate cell type proportions within each spot from referenced single-cell data. In our study we generated snRNA-seq data originating from the same tissues that were used in our spatial transcriptomics experiments. Only the snRNA-seq data generated in our study has the resolution to inform about the cell type composition within the analyzed tissue. The spatial transcriptomics data can not discern cell types but can inform about the proportions of cell types identified from a single-cell experiment across a tissue section based on a probabilistic method (stereoscope) that uses single cell data to deconvolve the cell mixtures in spatial data (2,3).

To clarify this distinction more prominently and avoid any confusion in our study we have revised the manuscript accordingly:

“Each tissue region that forms a data point in the ST analysis consists of a small mixture of cells. Therefore, we performed additional single-nuclei RNA-sequencing (snRNA-seq) to deconvolve spatial data and artificially increase the resolution in our analyses (Figure 1a).” (line 124- 127)

(1) <https://pubmed.ncbi.nlm.nih.gov/34857782/>

(2) <https://pubmed.ncbi.nlm.nih.gov/33037292/>

(3) https://docs.scvi-tools.org/en/stable/user_guide/models/stereoscope.html

First, we would like to highlight that we do not claim that the snSeq data and spatial data provide similar information, but rather deconvolve cell type information in spatial data, which indeed represents a limitation, and which we believe is stressed sufficiently in our manuscript between lines 100-103, 125-127, 282-284 and 429-431. However, to highlight this limitation even further we changed the manuscript text in the discussion between line 429-431, to explicitly mention the need for single-cell resolution for future adaptations of the method used by us to study *Plasmodium* liver infection: “To this end, high-resolution spatial omics applications, at cellular resolution, will be indispensable for understanding the coordination of immune priming in events of partial or full immunization.”

Specifically, in our study, we attempted to address this question by estimating cell type proportions across spatial positions from corresponding single nuclei data of the same tissue types. First, we would like to address the suggested lack of consistency between the spatial and snSeq data by the reviewer. We are uncertain which aspects the reviewer considers as inconsistent as they correctly point out that i) we retrieved the data from the same liver samples for both snSeq and ST experiments and ii) the estimation of cell type proportions from the snSeq reference we generated, inherently only considers genes which are present in both datasets used in our analysis. In detail, the method for single-cell deconvolution accounts for asymmetric data sets with flexible parameters that can adjust to the data. First, each cell type’s expression profile is characterized in the snSeq data and then the combination of these types that best explains the spatial data is estimated within each position (1). In case our explanation is insufficient to alleviate the reviewer from their doubts about the consistency of our data, it would be very helpful if the reviewer could specify the inconsistencies they are referring to further.

Next, regarding the differential expression of the genes shown in review figure 1 (in the previous rebuttal), we would like to refer the reviewer to figure 4 and the corresponding section of the manuscript between lines 279-339, which we dedicated to explaining changes in the expression of cells between the tested conditions in our data. As shown in Figure 4b, for a subset of immune cell types, which we considered of particular interest for infection, we do not observe differential cell type proportions between infected and control tissue. Based on this observation we continued to investigate whether gene expression of marker genes of these cell types differ between individual conditions, i.e. timepoint and infection status. For example, *Marco* which is considered a marker gene for Kupffer cells, together with a battery of additional Kupffer cell marker genes is upregulated in infected conditions, in particular at 12 and 24 hpi when compared to the controls

(Figure 4c, Supplementary data 4, lines 293-296). For the remaining genes shown in Review Figure 1 (*Itih3* and *Saa1*) we can infer proportions of cell types from the data provided. For example *Saa1* shows elevated expression in inflammatory hepatocytes (Review figure 1, Supplementary data 4, lines 311-313), which appear to be equally distributed across spatial clusters but with slight enrichment in proximity to parasite positions at 12 and 24 hpi (lines 309-311, Figure 5c). *Saa1* in our spatial data shows highest expression levels in ST3 and ST10 (Supplementary data 1, lines 183-184). Taken together, our observations suggest that differential expression levels of inflammatory hepatocytes are highly relevant during *P. berghei* infection in the liver. However, since our data describes transcriptomes in both, our snSeq and ST data, without defining the actual abundances (i.e. cell counts) of different cell types across the conditions in our study, we cannot

exclude with absolute certainty that cell type abundances play an equal or more crucial role during infection. However, we strongly anticipate that defining the absolute numbers of different cell types in our liver samples is beyond the scope of this study.

(1) <https://www.nature.com/articles/s42003-020-01247-y>